# Penalized Langevin dynamics with vanishing penalty for smooth and log-concave targets

**Avetik Karagulyan**
CREST, ENSAE, IP Paris
avetik.karagulyan@ensae.fr

**Arnak S. Dalalyan**
CREST, ENSAE, IP Paris
arnak.dalalyan@ensae.fr

## Abstract

We study the problem of sampling from a probability distribution on $\mathbb{R}^p$ defined via a convex and smooth potential function. We first consider a continuous-time diffusion-type process, termed Penalized Langevin dynamics (PLD), the drift of which is the negative gradient of the potential plus a linear penalty that vanishes when time goes to infinity. An upper bound on the Wasserstein-2 distance between the distribution of the PLD at time $t$ and the target is established. This upper bound highlights the influence of the speed of decay of the penalty on the accuracy of approximation. As a consequence, considering the low-temperature limit we infer a new nonasymptotic guarantee of convergence of the penalized gradient flow for the optimization problem.

## 1 Introduction

The problem of sampling from a probability distribution received a great deal of attention in machine learning literature. Gradient based MCMC methods such as the Langevin MC, the underdamped Langevin Monte Carlo, the Hamiltonian Monte Carlo and their Metropolis adjusted counterparts were shown to have attractive features both in practice and in theory. In particular, thanks to a large number of recent results, the case of smooth and strongly log-concave densities is now fairly well understood. In this case, non-asymptotic theoretical guarantees for various distances on probability distributions have been established, showing that the number of gradient evaluations necessary to achieve an error upper bounded by $\varepsilon$ is a low order polynomial of the dimension, the condition number and the inverse precision $1/\varepsilon$. The dependence on the latter is even logarithmic for Metropolis adjusted methods.

The main focus of this paper is on the problem of sampling from densities[1]

$$\pi(\boldsymbol{\theta}) \propto \exp(-f(\boldsymbol{\theta})), \qquad \boldsymbol{\theta} \in \mathbb{R}^p,$$

corresponding to a (weakly) convex potential function $f : \mathbb{R}^p \to \mathbb{R}$. In the sequel, a twice differentiable function $f : \mathbb{R}^p \to \mathbb{R}$ is said to satisfy $(m, M)$-SCGL condition ($m$-strongly convex and $M$-gradient Lipschitz), for some $M \geq m \geq 0$, if the following inequality is satisfied:

$$m\mathbf{I}_p \preceq \nabla^2 f(\boldsymbol{\theta}) \preceq M\mathbf{I}_p, \qquad \forall \boldsymbol{\theta} \in \mathbb{R}^p.$$

Here, for two square matrices $\mathbf{A}$ and $\mathbf{B}$, the relation $\mathbf{A} \preceq \mathbf{B}$ means that $\mathbf{B} - \mathbf{A}$ is positive semidefinite.

In this work, we wish to define a class of continuous-time processes, such that for every element $\{\boldsymbol{L}_t : t \geq 0\}$ of the class, the distribution of $\boldsymbol{L}$ at time $t$ is close to the target distribution $\pi$. When the potential function $f$ satisfies $(m, M)$-SCGL condition with $m \geq 0$, it is well-known that the vanilla

Langevin dynamics $L^{\mathsf{LD}}$, defined as the solution of

$$d\boldsymbol{L}_t^{\mathsf{LD}} = -\nabla f(\boldsymbol{L}_t^{\mathsf{LD}})dt + \sqrt{2}\,d\boldsymbol{W}_t, \tag{LD}$$

where $\boldsymbol{W}_t$ is a standard Wiener process independent of $\boldsymbol{L}_0^{\mathsf{LD}}$, has $\pi$ as invariant distribution (Bhattacharya, 1978). Furthermore, when $m > 0$, the distribution $\nu_t^{\mathsf{LD}}$ of $\boldsymbol{L}_t^{\mathsf{LD}}$ converges in Wasserstein distance (see below for a definition) exponentially fast to $\pi$ (Villani, 2008), that is

$$W_2(\nu_t^{\mathsf{LD}}, \pi) \le e^{-mt} W_2(\nu_0^{\mathsf{LD}}, \pi). \tag{1}$$

A remarkable feature of this result is that it is dimension free. In the case $m = 0$, it was established by (Bobkov, 1999) that the target distribution satisfies the Poincaré inequality with the Poincaré constant $\mathcal{C}_{\mathsf{P}}$ that might depend on the dimension. According to (Chewi et al., 2020), the exponentially fast convergence to zero holds true with $m$ replaced by $1/\mathcal{C}_{\mathsf{P}}$, when $m = 0$. In (Kannan et al., 1995), the authors conjectured that there is a universal constant $\mathcal{C}_{\mathsf{KLS}} > 0$ such that for any log-concave distribution $\pi$ on $\mathbb{R}^p$, $\mathcal{C}_{\mathsf{P}} \le \mathcal{C}_{\mathsf{KLS}} \|\mathbf{E}_{\boldsymbol{X} \sim \pi}[\boldsymbol{X}\boldsymbol{X}^\top]\|_{\mathrm{op}}$, where $\|\mathbf{A}\|_{\mathrm{op}}$ stands for the operator norm of the matrix $\mathbf{A}$. Despite important efforts made in recent years (see (Alonso-Gutiérrez and Bastero, 2015; Cattiaux and Guillin, 2018; Lee and Vempala, 2017)), this conjecture is still unproved. Note also that the Poincaré constant is, in general, hard to approximate and to estimate (Pillaud-Vivien et al., 2019). One approach to getting more tractable convergence bounds could be to find a tractable upper bound on the Poincaré constant of a distribution defined by a potential satisfying $(m, M)$-SCGL condition with $m = 0$. We develop here another approach, consisting in modifying the Langevin dynamics, so that the new dynamics has still a limiting distribution equal to $\pi$ but for which we can get a tractable upper bound. A natural way of defining this new dynamics is to add to the potential $f$ a strongly-convex penalty with a strong-convexity constant that vanishes when time goes to infinity. For a quadratic penalty function, this leads to the process $\boldsymbol{L}^{\mathsf{PLD}}$ termed penalized Langevin dynamics and defined by[2]

$$d\boldsymbol{L}_t^{\mathsf{PLD}} = -\big(\nabla f(\boldsymbol{L}_t^{\mathsf{PLD}}) + \alpha(t)\boldsymbol{L}_t^{\mathsf{PLD}}\big)\,dt + \sqrt{2}\,d\boldsymbol{W}_t, \tag{PLD}$$

where $\alpha : [0, \infty) \to [0, \infty)$ is a time-dependent penalty factor tending to zero as $t \to \infty$. The main result of this work is an upper bound on $W_2(\nu_t^{\mathsf{PLD}}, \pi)$ that is valid for every continuously differentiable and decreasing penalty factor $\alpha$. Optimizing over $\alpha$, we show that the choice $\alpha(t) \sim 1/(2t)$, when $t \to \infty$, leads to a simple upper bound of the order $1/\sqrt{t}$. Interestingly, using a suitably parametrized temperature-dependent potential function $f_\tau(\cdot) = f(\cdot)/\tau$ and a penalty factor, one can get an upper bound for the penalized gradient flow

$$\dot{\boldsymbol{X}}_t^{\mathsf{PGF}} = -\big(\nabla f(\boldsymbol{X}_t^{\mathsf{PGF}}) + \alpha_0(t)\boldsymbol{X}_t^{\mathsf{PGF}}\big), \qquad t \ge 0, \tag{PGF}$$

by passing to the low-temperature limit. This bound implies that $\|\boldsymbol{X}_t^{\mathsf{PGF}} - \boldsymbol{x}^*\|_2$ can be of the order $O(1/t^{1-\mathsf{q}})$, where $\boldsymbol{x}^*$ is a minimizer of the potential $f$ and $\mathsf{q} \in [0, 1]$ is a parameter appearing in an additional condition imposed on $f$. To the best of our knowledge, the obtained bound is new, most previous results being valid for the objective function itself, not for the minimum point.

The rest of the paper is organized as follows. We start by stating the bound on the error of the PLD in Section 2. We also instantiate the bound to the penalty factors that are inversely proportional to time. In Section 3, we discuss the connections with the optimization problem, assessing the error of the PGF. Section 4 is devoted to relation to prior work. The proof of the main result, up to some technical lemmas, is presented in Section 6. Missing proofs are deferred to the supplementary material.

To complete this introduction, we introduce some notations. We consider the Wasserstein-2 distance

$$W_2(\nu, \nu') = \inf\left\{\mathbf{E}[\|\boldsymbol{\vartheta} - \boldsymbol{\vartheta}'\|_2^2]^{1/2} : \boldsymbol{\vartheta} \sim \nu \text{ and } \boldsymbol{\vartheta}' \sim \nu'\right\},$$

where the minimum is over all joint distributions having $\nu$ and $\nu'$ as the first and the second marginal distributions. For any $\gamma > 0$, we define the probability density function $\pi_\gamma(\boldsymbol{\theta}) \propto \exp(-f(\boldsymbol{\theta}) - \gamma\|\boldsymbol{\theta}\|_2^2)$, where $\|\boldsymbol{\theta}\|_2$ is the Euclidean norm. We also define $\mu_k(\pi) = \mathbb{E}_{\boldsymbol{X} \sim \pi}[\|\boldsymbol{X}\|_2^k]$, the moment of order $k$ of $\pi$.

## 2 Convergence of penalized Langevin dynamics

In this section we explain our approach and state the main result. Without loss of generality, we will assume that the initial point for the PLD is the origin, $\boldsymbol{L}_0^{\mathsf{PLD}} = 0$. Note that if a good guess $\boldsymbol{\theta}_0$ of a minimizer of $f$ is available, it is recommended to initialize PLD at $\boldsymbol{\theta}_0$. Our framework covers this case, since it suffices to apply our results to the translated function $\widetilde{f}(\cdot) = f(\boldsymbol{\theta}_0 + \cdot)$. Under the condition $\boldsymbol{L}_0^{\mathsf{PLD}} = 0$, the Wasserstein-2 distance at the starting point coincides with the second-order moment, $W_2(\nu_0^{\mathsf{PLD}}, \pi) = \sqrt{\mu_2(\pi)}$.

When $\alpha(t) = \alpha$ is a strictly positive constant, the distribution $\nu_T^{\mathsf{PLD}}$, for a large value of $T$, is close to the biased target $\pi_\alpha$. Furthermore, in view of (1), the distance between these two distributions is smaller than a prescribed error level $\varepsilon > 0$ as soon as $T \geq (1/\alpha) \log(\sqrt{\mu_2(\pi)}/\varepsilon)$. On the other hand, one can choose $\alpha$ small enough such that the bias $W_2(\pi_\alpha, \pi)$ is smaller than $\varepsilon$. The discrete counterpart of this approach has been used in many recent works (Dalalyan, 2017; Dalalyan et al., 2019; Dwivedi et al., 2018). The approach we develop here extends these work to the case of time-dependent $\alpha$ and has the advantage of being asymptotically unbiased, when $t \to \infty$. In other words, it allows to choose $\alpha$ independently of $\varepsilon$ and make the error smaller than $\varepsilon$ by running PLD over a sufficiently large time period.

**Theorem 1.** *Suppose that $\pi$ is a probability distribution with a potential function $f$ that satisfies $(m, M)$-SCGL condition, where $m \geq 0$ and $M > 0$. Let $\alpha : [0, +\infty) \to \mathbb{R}$ be a non-increasing differentiable function, such that $m + \alpha(t) > 0$ for every $t \geq 0$. Then, for every positive number $t$ and for $\beta(t) = \int_0^t (m + \alpha(u)) \, du$, we have*

$$W_2(\nu_t^{\mathsf{PLD}}, \pi) \leq \sqrt{\mu_2(\pi)} \, e^{-\beta(t)} + 11 \mu_2(\pi) \left\{ e^{-\beta(t)} \int_0^t \frac{|\alpha'(s)| e^{\beta(s)}}{\sqrt{m + \alpha(s)}} \, ds + \frac{\alpha(t)}{\sqrt{m + \alpha(t)}} \right\}. \quad (2)$$

The proof being postponed to Section 6, the rest of this section is devoted to discussing the stated theorem and its consequences for some specific choices of the penalty factor. One can notice right away that in the case of a positive $m$, we can choose $\alpha$ to be zero, thereby obtaining the classical exponential convergence rate (Villani, 2008). In the rest of the discussion, we assume that $m = 0$.

The numerical constant 11 can certainly be improved. It is closely related to the fact that for a log-concave distribution $\nu$, we have $\mu_4(\nu) \leq C_4 \mu_2^2(\nu)$ for a universal constant $C_4$. proved to satisfy $C_4 \leq 442$ (Dalalyan et al., 2019, Remark 3). Improved bounds on $C_4$ will automatically yield improved numerical constant in Theorem 1. We also note that the Lipschitz constant $M$ does not appear in inequality (2). Our proof, however, requires the finiteness of $M$. We believe that it is possible to relax the gradient-Lipschitz assumption by requiring from $\nabla f$ to be only locally Lipschitz-continuous.

One can check that if we replace the penalty factor $\alpha$ by a larger function, the first term of the upper bound in (2), proportional to $\exp\{- \int_0^t \alpha(s) \, ds\}$, becomes smaller. On the other hand, the second term increases when $\alpha$ increases [3] One can thus use inequality (2) for choosing the penalty factor $\alpha$ that offers a trade-off between the error of approximating the biased density $\pi_{\alpha(t)}$ by the PLD and the error of approximating the target $\pi$ by the biased density $\pi_{\alpha(t)}$.

To "optimize" the upper bound with respect to $\alpha$, let us for the moment ignore the first term in the curly parentheses in (2). In that case our functional of interest has two components, where one of them is increasing with respect to $\alpha$, while the other is decreasing. (Here the monotony must be understood with a certain precaution, as in our case the mathematical concept is not well-defined.) These considerations suggest to choose the "optimal" $\alpha$ by balancing these two terms:

$$\sqrt{\mu_2(\pi)} \, e^{-\beta(t)} = 11 \sqrt{\alpha(t)} \, \mu_2(\pi). \quad (3)$$

Taking the square of both sides, cancelling out some terms and using that $\alpha(t) = \beta'(t)$, we check that (3) is equivalent to

$$121 \mu_2(\pi) \beta'(t) e^{2\beta(t)} = 1.$$

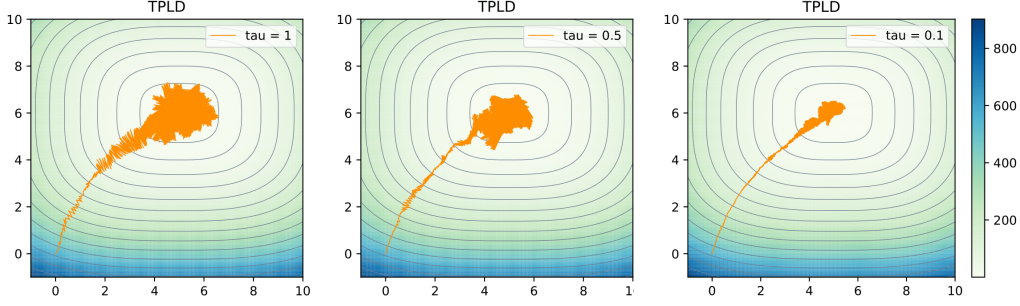

Figure 1: One random path of the TPLD, for $\tau = 1$ (left), $\tau = 0.5$ (middle) and $\tau = 0.1$ (right).

Solving this differential equation we get the following expression for $\beta(\cdot)$ and the corresponding expression for $\alpha(\cdot)$:

$$\beta^*(t) = \frac{1}{2}\log\left(\frac{2t}{121\mu_2(\pi)} + 1\right) \quad \text{and} \quad \alpha^*(t) = \frac{1}{2t + 121\mu_2(\pi)}.$$

It is easy to check that this choice of $\alpha$ ensures that the first and the last terms in the right hand side of (2) are of the order $1/\sqrt{t}$. Interestingly, the middle term in (2) turns out to be of the same order, up to a logarithmic factor. The precise statement of the consequence of Theorem 1 obtained by choosing $\alpha(t) = 1/(2t + A)$ for some $A > 0$ reads as follows.

**Proposition 1.** *If the potential function satisfies the $(m, M)$-SCGL condition with $m = 0$, then the error of PLD with $\alpha(t) = 1/(A + 2t)$, measured by the Wasserstein-2 distance, satisfies*

$$W_2(\nu_t^{\mathsf{PLD}}, \pi) \leq \frac{\sqrt{A\mu_2(\pi)} + 11\mu_2(\pi)\big(1 + \log\big(1 + (2/A)t\big)\big)}{\sqrt{A + 2t}}.$$

*In particular, for $A = 2\mu_2(\pi)$, we get*

$$W_2(\nu_t^{\mathsf{PLD}}, \pi) \leq \frac{10\mu_2(\pi)\big\{1 + \log\big(1 + (t/\mu_2(\pi))\big)\big\}}{\sqrt{\mu_2(\pi) + t}}.$$

To complete this section, we present a quick argument showing that the right hand side of (2) cannot converge to zero faster than the rate $1/\sqrt{t}$. Suppose that for a specific choice of $\alpha$, the right hand side of (2) is $o(1/t^{1/2})$. Then the last term is necessarily $o(1/t^{1/2})$, which implies that $\alpha(t) = o(1/t)$ when $t \to +\infty$. Thus, for some $c > 0$, $\alpha(t) \leq 1/(4t)$ for every $t \geq c$. This means that $\beta(t) \leq c\alpha(0) + (1/4)\log(t/c)$. Hence,

$$\exp(-\beta(t)) \geq (c/t)^{1/4}\exp(-c\alpha(0)).$$

This proves that the upper bound on the error of the PLD established in Theorem 1 may not tend to zero at a faster rate than $1/\sqrt{t}$, as $t$ goes to infinity. This argument also shows that the optimizer $\alpha(\cdot)$ of the upper bound is asymptotically equivalent to $1/(2t)$ when $t \to \infty$.

## 3 The counterpart in optimization: penalized gradient flow

In this section we draw the parallel between PLD and the penalized gradient flows, henceforth referred to as PGF, for a non-strongly convex function $f$. In the case of strongly convex functions, the gradient flow $\boldsymbol{X}_t^{\mathsf{GF}}$ defined by the differential equation $\dot{\boldsymbol{X}}_t^{\mathsf{GF}} = -\nabla f(\boldsymbol{X}_t^{\mathsf{GF}})$, converges exponentially fast to the minimum $\boldsymbol{x}_*$ of $f$, without the need to add a quadratic penalty. In contrast with this, for general non-strongly convex case functions $f$, only the convergence of the function $f(\boldsymbol{X}_t^{\mathsf{GF}})$ to $f(\boldsymbol{x}_*)$ at the rate $1/t$ can be established. The goal of this sections is to understand the convergence of the flow to the minimum point when a vanishing quadratic penalty is added to the cost function $f$; when does this flow converge, what is the impact of the penalty factor and what kind of rate can be achieved. To answers these questions, we assume in this section that $(0, M)$-SCGL holds true. We also assume that $f$ has a unique minimum point denoted by $\boldsymbol{x}_*$.

In the analysis performed in the previous section, we can replace the function $f(\cdot)$ by the function $f_\tau(\cdot) = f(\cdot)/\tau$. The function $f_\tau$ has $\boldsymbol{x}_*$ as its point of minimum, whatever the real number $\tau > 0$. Moreover, if we define the tempered density function $\pi^\tau(\boldsymbol{\theta}) \propto \exp\left(-f_\tau(\boldsymbol{\theta})\right)$, the distribution $\pi^\tau$ tends to $\delta_{\boldsymbol{x}_*}$, the Dirac mass at $\boldsymbol{x}_*$. Clearly, $f_\tau$ satisfies $(0, M/\tau)$-SCGL condition. Thus, according to Theorem 1, the process $\boldsymbol{L}_t^\tau$, defined as

$$\boldsymbol{L}_t^\tau = \boldsymbol{L}_0^\tau - \frac{1}{\tau}\int_0^t \left(\nabla f(\boldsymbol{L}_s^\tau) + \alpha(s/\tau)\boldsymbol{L}_s^\tau\right)ds + \sqrt{2}\,\boldsymbol{W}_t,$$

converges to $\pi^\tau$ in Wasserstein distance, if $\alpha(\cdot)$ is a continuously differentiable and non-increasing function. We now introduce the tempered penalized Langevin dynamics (TPLD), as a time-scaled version of $\boldsymbol{L}^\tau$: $\boldsymbol{X}_t^{\mathsf{TPLD}} = \boldsymbol{L}_{t\tau}^\tau$ for every $\tau > 0$. One can check that this process satisfies the stochastic differential equation

$$d\boldsymbol{X}_t^{\mathsf{TPLD}} = -\left(\nabla f(\boldsymbol{X}_t^{\mathsf{TPLD}}) + \alpha(t)\boldsymbol{X}_t^{\mathsf{TPLD}}\right)dt + \sqrt{2\tau}\,d\bar{\boldsymbol{W}}_t, \qquad \text{(TPLD)}$$

where $\bar{\boldsymbol{W}}_t = \tau^{-1/2}\boldsymbol{W}_{\tau t}$ is a standard Wiener process. To illustrate the behaviour of this process, Figure 1 shows one realization of TPLD for different values of $\tau$, with the left plot corresponding to PLD. All the results of the previous section continue to hold for this tempered dynamics. In particular, the analog of the second claim of Proposition 1 in the case of the tempered diffusion takes the following form.

**Proposition 2.** *If the potential function satisfies the* $(0, M)$-SCGL *condition, then the error of* TPLD *with* $\alpha(t) = 1/(2\mu_2(\pi^\tau) + 2t)$ *satisfies*

$$W_2(\nu_t^{\mathsf{TPLD}}, \pi^\tau) \leq \frac{10\mu_2(\pi^\tau)\left\{1 + \log\left(1 + (t/\mu_2(\pi^\tau))\right)\right\}}{\sqrt{\tau(\mu_2(\pi^\tau) + t)}}, \qquad \forall t \geq 0.$$

As mentioned above, for small $\tau$, $\pi^\tau$ is close to the Dirac mass at the minimum point $\boldsymbol{x}_*$. The last result tells us that we can approximate $\pi^\tau$ arbitrarily well, by running the TPLD over a large time-period. But we can not replace $\tau$ by zero in this result, since the denominator of the right hand side vanishes and the result becomes vacuous. We show below that this can be repaired if an additional assumption is introduced.

Taking $\tau = 0$ in (TPLD), we get the penalized gradient flow (PGF):

$$d\boldsymbol{X}_t^{\mathsf{PGF}} = -\left(\nabla f(\boldsymbol{X}_u^{\mathsf{PGF}}) + \alpha(u)\boldsymbol{X}_u^{\mathsf{PGF}}\right)du, \qquad t \geq 0, \qquad \boldsymbol{X}_0^{\mathsf{PGF}} = \boldsymbol{0}.$$

Here we recognize the analog of PLD in the setting of the gradient flows. On the other hand, the Euclidean distance on $\mathbb{R}^p$ is equal to the Wasserstein distance between Dirac measures. This leads us to think that our approach for obtaining non-asymptotic error bounds for PLD is applicable to the penalized gradient flow. This turns out to be true, modulo the introduction of the following assumption.

**Assumption** $\mathsf{A}(\mathsf{D}, \mathsf{q})$: The minimum point $\boldsymbol{x}_\gamma$ of the (strongly convex and coercive) function $f_\gamma(\cdot) = f(\cdot) + \gamma\|\cdot\|_2^2/2$ satisfies

$$\|\boldsymbol{x}_\gamma - \boldsymbol{x}_{\widetilde{\gamma}}\|_2 \leq \frac{\mathsf{D}}{\widetilde{\gamma}^{\mathsf{q}}}(\widetilde{\gamma} - \gamma)\|\boldsymbol{x}_*\|_2^{1-\mathsf{q}}, \qquad \forall \gamma < \widetilde{\gamma},$$

for some $\mathsf{D} > 0$ and $\mathsf{q} \in [0, 1]$.

Since $\boldsymbol{x}_\gamma$ a stationary point of $f_\gamma$, we have $\nabla f(\boldsymbol{x}_\gamma) = -\gamma\boldsymbol{x}_\gamma$. From this relation and (Nesterov, 2004, Theorem 2.1.12), one can deduce that (a) if $f$ satisfies $m$-strongly convex, then $\mathbf{A}(1, 1)$ holds and (b) if $f$ satisfies $(0, M)$-SCGL condition then $\mathbf{A}(M, 0)$ holds.

**Theorem 2.** *Assume that* $\alpha : [0, \infty) \to [0, \infty)$ *is a continuously differentiable and non-increasing function. Let* $\beta(t) = \int_0^t \alpha(s)\,ds$ *be the antiderivative of* $\alpha$. *If* $f$ *satisfies* $\mathbf{A}(\mathsf{D}, \mathsf{q})$ *and* $(0, M)$-SCGL, *then*

$$\|\boldsymbol{X}_t^{\mathsf{PGF}} - \boldsymbol{x}_*\|_2 \leq \|\boldsymbol{x}_*\|_2^{1-\mathsf{q}}\left(e^{-\beta(t)} + \mathsf{D}\int_0^t \frac{|\alpha'(s)|}{\alpha^{\mathsf{q}}(s)}e^{\beta(s)-\beta(t)}ds + \mathsf{D}\alpha^{1-\mathsf{q}}(t)\right). \qquad (4)$$

The proof can be found in Appendix D of the supplementary material. When $\mathsf{q} = 1/2$, this result is the optimization counterpart of the inequality shown in Theorem 1. Once again, it is appealing

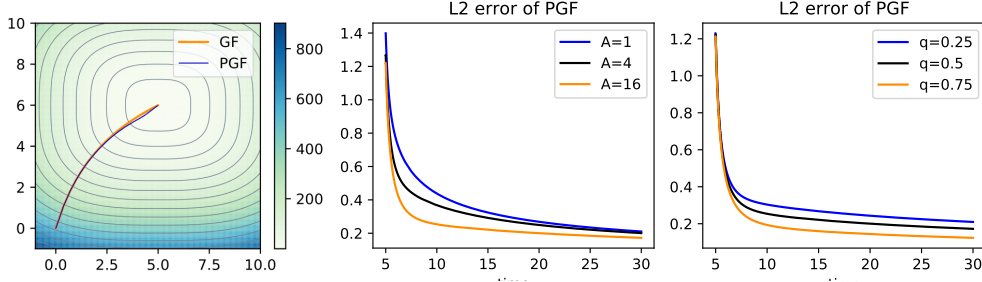

Figure 2: Left: The gradient flow (orange) and the penalized gradient flow (blue). Middle and Right: the mapping $t \mapsto \|\boldsymbol{X}_t^{\mathsf{PGF}} - \boldsymbol{x}_*\|_2$. In both plots we used the function $f(x, y) = |x - 5|^3 + 2|y - 6|^3$.

to optimize the right hand side of (4) in order to choose the "best" penalty factor. Using arguments similar to those of previous section, *i.e.*, balancing the first and the last terms on the right hand side of (4), we get that the "optimal" convergence of $\mathsf{PGF}$ is achieved when

$$\alpha^*(t) = \frac{(1 - \mathsf{q})}{t + \mathsf{D}^{1/(1-\mathsf{q})}(1 - \mathsf{q})} \qquad \text{and} \qquad \beta^*(t) = (1 - \mathsf{q}) \log\left(\frac{t}{\mathsf{D}^{1/(1-\mathsf{q})}(1 - \mathsf{q})} + 1\right).$$

This leads us to make the recommendation of choosing $\alpha(t) = (1 - \mathsf{q})/(t + A)$, for some positive $A$. If $\mathsf{q} = 1$, this amounts to considering the non-penalized gradient flow and (4) boils down to the fact that the distance from the gradient flow of a convex function to its minimum decreases. While for $\mathsf{q} < 1$, for the foregoing choice of the penalty factor, we get the error bound

$$\|\boldsymbol{X}_t^{\mathsf{PGF}} - \boldsymbol{x}_*\|_2 \leq \frac{A^{1-\mathsf{q}} + \mathsf{D} + \mathsf{D}\log(1 + (t/A))}{(t + A)^{1-\mathsf{q}}}\|\boldsymbol{x}_*\|_2. \tag{5}$$

To complete this section, we make some remarks on assumption $\mathbf{A}(\mathsf{D}, \mathsf{q})$. First, one can relax this assumption by requiring that the desired inequality holds for sufficiently small values of $\widetilde{\gamma}$ only. In this form, it can be easily seen that larger values of $\mathsf{q}$ correspond to weaker assumption. Second, even if the function $f$ is not strongly convex, it may satisfy $\mathbf{A}(\mathsf{D}, 0)$. An example is the function $f(x) = \sqrt{(x - x_*)^2 + b^2}$. The second derivative of this function is equal to $b^2/((x - x_*)^2 + b^2)^{3/2}$. This implies that $f$ satisfies $(0, 1/b)$-$\mathsf{SCGL}$. We show in the supplementary material that it satisfies $\mathbf{A}(\mathsf{D}, 0)$ for some finite value of $\mathsf{D} > 0$. This is not really surprising, given that this function is strongly convex on any compact set. Another instructive example is the function $f(x) = |x - x_*|^a$ with $a \geq 2$. On compact sets, this function satisfies[4] $\mathbf{A}(\mathsf{D}, (a - 2)/(a - 1))$. Therefore, the error bound (5) implies that $\mathsf{PGF}$ converges to the minimum of $f$ at the rate $1/t^{1/(a-1)}$, which is faster than the rate derived from the standard $O(1/t)$ bound for the non-penalized gradient flow. This behaviour is depicted in Figure 2.

## 4  Prior work and outlook

The relation of our results to some prior work has already been highlighted in previous sections. This section provides some complementary bibliographic remarks on recent advances on Langevin diffusions, gradient flows and their discrete counterparts.

Convergence of Langevin dynamics in continuous time has received a lot of attention in probability, see (Cattiaux and Guillin, 2009; Cattiaux et al., 2010; Bolley et al., 2012) and the references therein. An interesting known fact, for instance, is that the Langevin dynamics satisfies[5] $W_2(\nu_{t,\boldsymbol{x}}^{\mathsf{LD}}, \nu_{t,\boldsymbol{y}}^{\mathsf{LD}}) \leq e^{-mt}\|\boldsymbol{x} - \boldsymbol{y}\|_2$ if and only if $f$ is $m$-strongly convex. More recently, many papers in statistics and machine learning literature established non-asymptotic error bounds for discretized algorithms, mainly focusing on the convex case, see (Durmus and Moulines, 2019, 2017; Hsieh et al., 2018; Bubeck et al., 2018; Shen and Lee, 2019) in addition to previously cited papers. The non-convex

case was emphasized in (Cheng et al., 2018; Majka et al., 2018; Erdogdu et al., 2018; Mangoubi and Vishnoi, 2019).

In the optimization setting, the results on the convergence of the gradient flow for convex objectives have been known for a long time. More recently, (Su et al., 2016) derived a continuous-time second-order differential equation characterizing the Nesterov acceleration. Continuous-time Accelerated Mirror Descent was studied in (Krichene et al., 2015). An approach based on Bregman-Lagrangian functional for continuous-time momentum and other methods was developed in (Wibisono et al., 2016; Wilson et al., 2016). Further results on related topics, relevant to machine learning, were obtained in (Zhang et al., 2018; Scieur et al., 2017; Franca et al., 2018). An overview of results on gradient flow beyond the Euclidean space setting can be found in (Ambrosio et al., 2008; Santambrogio, 2017).

On a related note, several studies took advantage of the fact that the distribution of the Langevin dynamics is a gradient flow in the space of measures (Cheng and Bartlett, 2018; Bernton, 2018; Durmus et al., 2018a; Wibisono, 2018), in order to establish error bounds for sampling algorithms. The relation with MMD was studied by Arbel et al. (2019).

The results presented in the present work can be generalized in various directions. In particular, it would be interesting to relax the smoothness assumption, following an argument from, *e.g.*, (Durmus et al., 2018b; Chatterji et al., 2019; Mou et al., 2019; Salim et al., 2019), to develop a similar theory for the kinetic Langevin dynamics (Eberle et al., 2019; Cheng et al., 2017; Dalalyan and Riou-Durand, 2018; Ma et al., 2019) or to consider the case of a Lévy process driven stochastic differential equation in the spirit of (Simsekli et al., 2020; Liang et al., 2019).

## 5   Conclusion

We put forward a family of time-inhomogeneous diffusion processes that converge to a pre-specified target distribution and, therefore, can be used for approximate sampling. These processes are defined as penalized Langevin dynamics with a penalty that vanishes when time goes to infinity. The penalty allows to ensure strong convexity, which helps to handle situations where the original log-density is not strongly convex. We established a simple non-asymptotic error bound showing that the rate of convergence in the Wasserstein-2 distance is $O(1/\sqrt{t})$. We have also discussed analogous results for the penalized gradient flow. The important next step to investigate in future works is the analysis of discretized versions of the penalized Langevin dynamics.

## 6   Proof of Theorem 1

Recall that for every $\gamma \in \mathbb{R}$, $\pi_\gamma$ is the probability distribution with density proportional to $\exp(-f(\boldsymbol{\theta}) - \gamma\|\boldsymbol{\theta}\|_2^2/2)$. The triangle inequality for the Wasserstein distance yields

$$W_2(\nu_t^{\mathsf{PLD}}, \pi) \leq W_2(\nu_t^{\mathsf{PLD}}, \pi_{\alpha(t)}) + W_2(\pi_{\alpha(t)}, \pi), \tag{6}$$

for every $t > 0$. We will bound these two terms separately, but let us start by stating two technical lemmas. The first one is a consequence of the well-known transportation cost inequality (see (Gozlan and Léonard, 2010, Corollary 7.2)), whereas the second one establishes the smoothness and the monotony with respect to $\gamma$ of the second-order moment of $\pi_\gamma$. The proofs of these lemmas are postponed to Appendix A.

**Lemma 1.** *Let $\pi$ be a probability density function such that the potential $f = -\log(\pi)$ satisfies the $(m, +\infty)$-$\mathsf{SCGL}$ condition. Let $\widetilde{\gamma} \geq \gamma$ be real numbers, such that $m + \gamma \geq 0$. Then*

$$W_2(\pi_{\widetilde{\gamma}}, \pi_\gamma) \leq \frac{11(\widetilde{\gamma} - \gamma)}{\sqrt{m + \widetilde{\gamma}}}\mu_2(\pi_\gamma).$$

**Lemma 2.** *Suppose that $\pi$ has a finite fourth-order moment. Then $\gamma \mapsto \mu_2(\pi_\gamma)$ is continuously differentiable and non-increasing, when $\gamma \in [0, +\infty)$.*

If we apply Lemma 1 with $\gamma = 0$ and $\widetilde{\gamma} = \alpha(t)$, then we obtain

$$W_2(\pi_{\alpha(t)}, \pi) \leq \frac{11\alpha(t)}{\sqrt{m + \alpha(t)}}\mu_2(\pi). \tag{7}$$

This provides the desired upper bound on the second term of the right hand side of (6). To bound the first term, $W_2(\nu_t^{\mathsf{PLD}}, \pi_{\alpha(t)})$, we aim at obtaining a Gronwall-type inequality for the function

$$\phi(t) := W_2(\nu_t^{\mathsf{PLD}}, \pi_{\alpha(t)}).$$

To this end, we consider an auxiliary stochastic process $\{\widetilde{\boldsymbol{L}}_u : u \geq t\}$, defined as a solution of the following stochastic differential equation

$$d\widetilde{\boldsymbol{L}}_u = -\big(\nabla f(\widetilde{\boldsymbol{L}}_u) + \alpha(t)\widetilde{\boldsymbol{L}}_u\big)du + \sqrt{2}d\boldsymbol{W}_u,$$

with the starting point $\widetilde{\boldsymbol{L}}_t = \boldsymbol{L}_t^{\mathsf{PLD}}$. This is in fact the Langevin diffusion corresponding to the potential $f(\cdot) + \alpha(t)\|\cdot\|_2^2/2$. Therefore, $\pi_{\alpha(t)}$ is the invariant distribution of $\widetilde{\boldsymbol{L}}$, and it is $(m + \alpha(t))$-strongly log-concave. Let $\boldsymbol{Q}_{t,\delta}$ be the distribution of the random vector $\widetilde{\boldsymbol{L}}_{t+\delta}$. The triangle inequality yields

$$\phi(t + \delta) \leq W_2\big(\nu_{t+\delta}^{\mathsf{PLD}}, \boldsymbol{Q}_{t,\delta}\big) + W_2\big(\boldsymbol{Q}_{t,\delta}, \pi_{\alpha(t)}\big) + W_2\big(\pi_{\alpha(t)}, \pi_{\alpha(t+\delta)}\big).$$

Recalling the definition of $\pi_{\alpha(t)}$ and $\boldsymbol{Q}_{t,\delta}$, we therefore find ourselves in the case of classical Langevin diffusion. Hence, one can apply (1) to get the bound

$$W_2(\boldsymbol{Q}_{t,\delta}, \pi_{\alpha(t)}) \leq \exp\big(-\delta(m + \alpha(t))\big)W_2(\nu_t^{\mathsf{PLD}}, \pi_{\alpha(t)}) = \exp\big(-\delta(m + \alpha(t))\big)\phi(t).$$

Applying Lemma 1 to $\pi_{\alpha(t)}$ and $\pi_{\alpha(t+\delta)}$, we get

$$W_2\big(\pi_{\alpha(t)}, \pi_{\alpha(t+\delta)}\big) \leq \frac{11(\alpha(t) - \alpha(t+\delta))}{\sqrt{m + \alpha(t)}}\mu_2(\pi_{\alpha(t+\delta)}).$$

Thus we obtain a bound for $\phi(t + \delta)$, that depends linearly on $\phi(t)$:

$$\phi(t + \delta) \leq W_2\big(\nu_{t+\delta}^{\mathsf{PLD}}, \boldsymbol{Q}_{t,\delta}\big) + e^{-\delta(m+\alpha(t))}\phi(t) + \frac{11(\alpha(t) - \alpha(t+\delta))}{\sqrt{m + \alpha(t)}}\mu_2(\pi_{\alpha(t+\delta)}). \quad (8)$$

Let us subtract $\phi(t)$ from both sides of (8) and divide by $\delta$:

$$\frac{\phi(t + \delta) - \phi(t)}{\delta} \leq \frac{1}{\delta} \cdot W_2\big(\nu_{t+\delta}^{\mathsf{PLD}}, \boldsymbol{Q}_{t,\delta}\big) + \frac{\exp\big(-\delta(m + \alpha(t))\big) - 1}{\delta} \cdot \phi(t)$$
$$+ \frac{11(\alpha(t) - \alpha(t+\delta))}{\delta\sqrt{m + \alpha(t)}}\mu_2(\pi_{\alpha(t+\delta)}). \quad (9)$$

The next lemma provides an upper bound on $W_2\big(\nu_{t+\delta}^{\mathsf{PLD}}, \boldsymbol{Q}_{t,\delta}\big)$ showing that it is $o(\delta)$, when $\delta \to 0$.

**Lemma 3.** *For every $t, \delta > 0$, and for every integrable function $\alpha : [t, t + \delta] \to [0, \infty)$,*

$$W_2\big(\nu_{t+\delta}^{\mathsf{PLD}}, \boldsymbol{Q}_{t,\delta}\big) \leq \big(\phi(t) + \mu_2^{1/2}(\pi)\big)\exp\left\{M\delta + \int_0^\delta \alpha(t+u)\,du\right\}\int_0^\delta \big|\alpha(t+s) - \alpha(t)\big|\,ds.$$

When $\delta$ tends to 0, according to Lemma 3, the first term of the right-hand side of (9) vanishes. Thus, after passing to the limit, we are left with the following Gronwall-type inequality:

$$\phi'(t) \leq -(m + \alpha(t))\phi(t) - \frac{11\alpha'(t)}{\sqrt{m + \alpha(t)}} \cdot \mu_2(\pi_{\alpha(t)}). \quad (10)$$

Here we tacitly used the fact that $\mu_2(\pi_{\alpha(t+\delta)}) \to \mu_2(\pi_{\alpha(t)})$, whenever $\delta \to 0$, which is due to the continuity of $\alpha(t)$ and Lemma 2. Recalling that the function $\beta(t)$ is given by $\beta(t) = \int_0^t \big(m + \alpha(s)\big)\,ds$, one can rewrite (10) as

$$\big(\phi(t)e^{\beta(t)}\big)' \leq -\frac{11\alpha'(t)e^{\beta(t)}}{\sqrt{m + \alpha(t)}}\mu_2(\pi_{\alpha(t)}) \leq -\frac{11\alpha'(t)e^{\beta(t)}}{\sqrt{m + \alpha(t)}}\mu_2(\pi).$$

Therefore we infer the following bound on $\phi(t)$:

$$\phi(t) \leq \phi(0)e^{-\beta(t)} - 11\mu_2(\pi)\int_0^t \frac{\alpha'(s)}{\sqrt{m + \alpha(s)}}e^{\beta(s)-\beta(t)}ds.$$

Combining this bound with (6) and (7), we obtain the inequality

$$W_2(\nu_t^{\mathsf{PLD}}, \pi) \leq W_2(\nu_0, \pi_{\alpha(0)})e^{-\beta(t)} - 11\mu_2(\pi)\int_0^t \frac{\alpha'(s)e^{\beta(s)-\beta(t)}}{\sqrt{m + \alpha(s)}}ds + \frac{11\alpha(t)\mu_2(\pi)}{\sqrt{m + \alpha(t)}}.$$

Lemma 2 yields $W_2(\nu_0, \pi_{\alpha(0)}) = \sqrt{\mu_2(\pi_{\alpha(0)})} \leq \sqrt{\mu_2(\pi)}$. This completes the proof of Theorem 1, since the derivative of $\alpha$ is negative.

## 7 Broader impact

We have developed a mathematical framework, that allows us to give quantitative non-asymptotic results for approximate sampling from non-strongly log-concave distributions. The main idea lies in adjusting the well-known Langevin SDE, in a way that the solution of the new equation converges to the target distribution. In this paper we have done the analysis of the continuous process. As a continuation of our work, one can extend it for the case of discrete-time processes, which can later be used in applications in Machine Learning and Computer Vision, where sampling from high-dimensional distributions is often required. This paper is purely theoretical, thus we expect no direct or indirect ethical risks from it. Nevertheless the described results provide a solid ground for future works that can directly be applied to tackle real-world problems.

## Footnotes

[1]We will use the same notation for the probability density functions and corresponding distributions.

[2]If a good initial guess $\boldsymbol{\theta}_0$ of the minimum point of $f$ is available, it might be better to replace the penalty term by $\alpha(t)(\cdot - \boldsymbol{\theta}_0)$.

[3]This is clear for the last term, proportional to $\sqrt{\alpha(t)}$, whereas the corresponding claim for the first term in the curly parentheses less trivial.

[4]See supplementary material.

[5]Here, $\nu_{t,\boldsymbol{x}}^{\mathsf{LD}}$ is the distribution at time $t$ of the $\mathsf{LD}$ starting at $\boldsymbol{x}$, and the inequality is assumed to hold for any $\boldsymbol{x}, \boldsymbol{y} \in \mathbb{R}^p$ and any $t \in \mathbb{R}$.

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
