[Supplementary Material]

# Appendix

## Table of Contents

## A  Proofs of the lemmas used in Theorem 1

### A.1  Proof of Lemma 1

We denote by $D_{\mathrm{KL}}(\pi_\gamma || \pi_{\widetilde{\gamma}})$ the Kullback-Leibler divergence between the distributions $\pi_\gamma$ and $\pi_{\widetilde{\gamma}}$. Since $\pi_{\widetilde{\gamma}}$ is $(m + \widetilde{\gamma})$-strongly log-concave, the transportation cost inequality (Gozlan and Léonard, 2010, Corollary 7.2) yields

$$W_2^2(\pi_{\widetilde{\gamma}}, \pi_\gamma) \leq \frac{2}{m + \widetilde{\gamma}} D_{\mathrm{KL}}(\pi_\gamma || \pi_{\widetilde{\gamma}}). \tag{11}$$

Let us denote by $c_\gamma$ the logarithm of the normalizing constant for $\pi_\gamma$ so that $\pi_\gamma(\boldsymbol{\theta}) = \exp(-f(\boldsymbol{\theta}) - (1/2)\gamma \|\boldsymbol{\theta}\|_2^2 + c_\gamma)$. Similarly, we denote by $c_{\widetilde{\gamma}}$ the logarithm of the normalizing constant of $\pi_{\widetilde{\gamma}}$. This readily yields

$$\begin{aligned}
D_{\mathrm{KL}}(\pi_\gamma || \pi_{\widetilde{\gamma}}) &= \int_{\mathbb{R}^p} \pi_\gamma(\boldsymbol{\theta}) \log \left( \frac{\pi_\gamma(\boldsymbol{\theta})}{\pi_{\widetilde{\gamma}}(\boldsymbol{\theta})} \right) d\boldsymbol{\theta} \\
&= \int_{\mathbb{R}^p} \pi_\gamma(\boldsymbol{\theta}) \left( 1/2(\widetilde{\gamma} - \gamma) \|\boldsymbol{\theta}\|_2^2 + c_\gamma - c_{\widetilde{\gamma}} \right) d\boldsymbol{\theta} \\
&= 1/2(\widetilde{\gamma} - \gamma) \mu_2(\pi_\gamma) + c_\gamma - c_{\widetilde{\gamma}}.
\end{aligned}$$

Using the inequality $e^{-x} \leq 1 - x + (1/2)x^2$ for all $x > 0$ implies the following upper bound on $c_\gamma - c_{\widetilde{\gamma}}$:

$$\begin{aligned}
c_\gamma - c_{\widetilde{\gamma}} &= \log \left( \int_{\mathbb{R}^p} \pi_\gamma(\boldsymbol{\theta}) \exp \left( 1/2(\gamma - \widetilde{\gamma}) \|\boldsymbol{\theta}\|_2^2 \right) d\boldsymbol{\theta} \right) \\
&\leq \log \left( 1 + 1/2(\gamma - \widetilde{\gamma}) \mu_2(\pi_\gamma) + 1/8(\gamma - \widetilde{\gamma})^2 \mu_4(\pi_\gamma) \right).
\end{aligned}$$

Since $\log(1 + x) \leq x$ for $x > -1$ we get

$$D_{\mathrm{KL}}(\pi_\gamma || \pi_{\widetilde{\gamma}}) \leq 1/8(\gamma - \widetilde{\gamma})^2 \mu_4(\pi_\gamma).$$

Since $m + \gamma \geq 0$, the distribution $\pi_\gamma$ is log-concave. Thus, in view of (Dalalyan et al., 2019, Remark 3), we have the inequality $\mu_4(\pi_\gamma) \leq 442\mu_2^2(\pi_\gamma)$. Finally, combining these bounds with (11), we get

$$W_2(\pi_{\widetilde{\gamma}}, \pi_\gamma) \leq \sqrt{\frac{2}{m + \widetilde{\gamma}}} \times \frac{(\widetilde{\gamma} - \gamma)\mu_4^{1/2}(\pi_\gamma)}{\sqrt{8}} \leq \frac{11\mu_2(\pi_\gamma)}{\sqrt{m + \widetilde{\gamma}}}(\widetilde{\gamma} - \gamma).$$

This completes the proof of the lemma.

## A.2   Proof of Lemma 2

For $k \in \mathbb{N} \cup \{0\}$, define

$$h_k(\gamma) = \int_{\mathbb{R}^p} \|\boldsymbol{\theta}\|_2^k \exp\left(-f(\boldsymbol{\theta}) - \gamma\|\boldsymbol{\theta}\|_2^2/2\right) d\boldsymbol{\theta}.$$

If $\pi \in \mathcal{P}_k(\mathbb{R}^p)$ then the function $h_k$ is continuous on $[0; +\infty)$. Indeed, if the sequence $\{\gamma_n\}_n$ converges $\gamma_0$, when $n \to +\infty$, then the function $\|\boldsymbol{\theta}\|_2^k \exp\left(-f(\boldsymbol{\theta}) - (1/2)\gamma_n\|\boldsymbol{\theta}\|_2^2\right)$ is upper-bounded by $\|\boldsymbol{\theta}\|_2^k \exp\left(-f(\boldsymbol{\theta})\right)$. Thus in view of the dominated convergence theorem, we can interchange the limit and the integral. Since, by definition,

$$\mu_k(\pi_\gamma) = \frac{h_k(\gamma)}{h_0(\gamma)},$$

we get the continuity of $\mu_2(\pi_\gamma)$ and $\mu_4(\pi_\gamma)$. Let us now prove that $h_k(t)$ is continuously differentiable, when $\pi \in \mathcal{P}_{k+2}(\mathbb{R}^p)$. The integrand function in the definition of $h_k$ is a continuously differentiable function with respect to $t$. In addition, its derivative is continuous and is as well integrable on $\mathbb{R}^p$, as we supposed that $\pi$ has the $(k+2)$-th moment. Therefore, Leibniz integral rule yields the following

$$h_k'(\gamma) = -\frac{1}{2}\int_{\mathbb{R}^p} \|\boldsymbol{\theta}\|_2^{k+2} \exp\left(-f(\boldsymbol{\theta}) - \gamma\|\boldsymbol{\theta}\|_2^2/2\right) d\boldsymbol{\theta} = -\frac{1}{2}h_{k+2}(t).$$

The latter yields the smoothness of $h_k$. Finally, in order to prove the monotony of $\mu_2(\pi_\gamma)$, we will simply calculate its derivative

$$\begin{aligned}
(\mu_2(\pi_\gamma))' &= -\frac{1}{2h_0(\gamma)}h_4(\gamma) - \frac{h_0'(\gamma)}{h_0(\gamma)^2}h_2(\gamma) \\
&= -\frac{1}{2}\mu_4(\pi_\gamma) + \frac{h_2^2(\gamma)}{2h_0(\gamma)^2} \\
&= \frac{1}{2}\left(\mu_2^2(\pi_\gamma) - \mu_4(\pi_\gamma)\right).
\end{aligned}$$

Since the latter is always negative, this completes the proof of the lemma.

## A.3   Proof of Lemma 3

From the definition of Wasserstein distance, we have

$$W_2\left(\nu_{t+\delta}^{\mathsf{PLD}}, \boldsymbol{Q}_{t,\delta}\right) \leq \|\widetilde{\boldsymbol{L}}_{t+\delta} - \boldsymbol{L}_{t+\delta}^{\mathsf{PLD}}\|_{\mathbb{L}_2}.$$

In view of the definition of the process $\widetilde{\boldsymbol{L}}$, we can write

$$\widetilde{\boldsymbol{L}}_{t+\delta} - \boldsymbol{L}_{t+\delta}^{\mathsf{PLD}} = \int_t^{t+\delta} \left(\nabla f(\boldsymbol{L}_s^{\mathsf{PLD}}) - \nabla f(\widetilde{\boldsymbol{L}}_s) + \alpha(s)\boldsymbol{L}_s^{\mathsf{PLD}} - \alpha(t)\widetilde{\boldsymbol{L}}_s\right) ds.$$

Therefore we have

$$\|\widetilde{\boldsymbol{L}}_{t+\delta} - \boldsymbol{L}_{t+\delta}^{\mathsf{PLD}}\|_{\mathbb{L}_2} \leq \underbrace{\left\| \int_t^{t+\delta} \left(\nabla f(\boldsymbol{L}_s^{\mathsf{PLD}}) - \nabla f(\widetilde{\boldsymbol{L}}_s)\right) ds \right\|_{\mathbb{L}_2}}_{:=T_1} + \underbrace{\left\| \int_t^{t+\delta} \left(\alpha(s)\boldsymbol{L}_s^{\mathsf{PLD}} - \alpha(t)\widetilde{\boldsymbol{L}}_s\right) ds \right\|_{\mathbb{L}_2}}_{:=T_2}.$$

Now let us analyze these two terms separately. We start with $T_1$:

$$\|T_1\|_{\mathbb{L}_2} = \left\| \int_t^{t+\delta} \left( \nabla f(\boldsymbol{L}_s^{\mathsf{PLD}}) - \nabla f(\widetilde{\boldsymbol{L}}_s) \right) ds \right\|_{\mathbb{L}_2}$$

$$\leq \int_t^{t+\delta} \left\| \nabla f(\boldsymbol{L}_s^{\mathsf{PLD}}) - \nabla f(\widetilde{\boldsymbol{L}}_s) \right\|_{\mathbb{L}_2} ds$$

$$\leq M \int_t^{t+\delta} \| \boldsymbol{L}_s^{\mathsf{PLD}} - \widetilde{\boldsymbol{L}}_s \|_{\mathbb{L}_2} ds.$$

These are due to the Minkowskii inequality and the Lipschitz continuity of the gradient. In order to bound the second term $T_2$, we will add and subtract the term $\alpha(t+s)\widetilde{\boldsymbol{L}}_{t+s}$. Similar to the case above, we get the following upper bound:

$$\|T_2\|_{\mathbb{L}_2} \leq \int_t^{t+\delta} \alpha(s) \left\| \boldsymbol{L}_s^{\mathsf{PLD}} - \widetilde{\boldsymbol{L}}_s \right\|_{\mathbb{L}_2} ds + \int_t^{t+\delta} |\alpha(s) - \alpha(t)| \left\| \widetilde{\boldsymbol{L}}_s \right\|_{\mathbb{L}_2} ds$$

$$= \int_0^{\delta} \alpha(t+s) \left\| \boldsymbol{L}_{t+s}^{\mathsf{PLD}} - \widetilde{\boldsymbol{L}}_{t+s} \right\|_{\mathbb{L}_2} ds + \int_0^{\delta} |\alpha(t+s) - \alpha(t)| \left\| \widetilde{\boldsymbol{L}}_{t+s} \right\|_{\mathbb{L}_2} ds.$$

Recall that $\widetilde{\boldsymbol{L}}_{t+s}$ is the solution of Langevin SDE with an $(m + \alpha(t))$-strongly convex potential function, and $Q_{t,s}$ is its distribution on $\mathbb{R}^p$. Thus, the triangle inequality yields

$$\left\| \widetilde{\boldsymbol{L}}_{t+s} \right\|_{\mathbb{L}_2} = W_2(\boldsymbol{Q}_{t,s}, \delta_0) \leq W_2(\boldsymbol{Q}_{t,s}, \pi_{\alpha(t)}) + W_2(\pi_{\alpha(t)}, \delta_0)$$

$$\leq W_2(\nu_t^{\mathsf{PLD}}, \pi_{\alpha(t)}) \exp(-ms - \alpha(t)s) + \sqrt{\mu_2(\pi_{\alpha(t)})}$$

$$\leq W_2(\nu_t^{\mathsf{PLD}}, \pi_{\alpha(t)}) + \sqrt{\mu_2(\pi_{\alpha(t)})} := V_t.$$

Summing up, we have

$$\left\| \boldsymbol{L}_{t+\delta}^{\mathsf{PLD}} - \widetilde{\boldsymbol{L}}_{t+\delta} \right\|_{\mathbb{L}_2} \leq \int_0^{\delta} \left( M + \alpha(t+s) \right) \| \boldsymbol{L}_{t+s}^{\mathsf{PLD}} - \widetilde{\boldsymbol{L}}_{t+s} \|_{\mathbb{L}_2} ds + \widetilde{\alpha}_t(\delta)\, V_t,$$

where $\widetilde{\alpha}_t(\delta)$ is an auxiliary function defined as

$$\widetilde{\alpha}_t(\delta) := \int_0^{\delta} |\alpha(t+s) - \alpha(t)|\, ds.$$

Now let us define $\Phi(s) = \| \boldsymbol{L}_{t+s}^{\mathsf{PLD}} - \widetilde{\boldsymbol{L}}_{t+s} \|_{\mathbb{L}_2}$. The last inequality can be rewritten as

$$\Phi(\delta) \leq \int_0^{\delta} \left( M + \alpha(t+s) \right) \Phi(s)\, ds + \widetilde{\alpha}_t(\delta)\, V_t.$$

The (integral form of the) Gronwall inequality, lemma 5, implies that

$$\Phi(\delta) \leq V_t \int_0^{\delta} \widetilde{\alpha}_t(s) \left( M + \alpha(t+s) \right) e^{\int_s^{\delta} (M + \alpha(t+u))\, du}\, ds + \widetilde{\alpha}_t(\delta) V_t$$

$$= V_t \int_0^{\delta} \widetilde{\alpha}_t'(s)\, e^{\int_s^{\delta} (M + \alpha(t+u))\, du}\, ds$$

$$\leq V_t\, \widetilde{\alpha}_t(\delta)\, \exp \left\{ M\delta + \int_0^{\delta} \alpha(t+u)\, du \right\}.$$

This completes the proof.

### A.4 Different forms of the Gronwall inequality

In this section, we provide two forms of the Gronwall inequality that are used in the present work. For the sake of the self-containedness, the proofs of these inequalities are also provided.

**Lemma 4** (Differential form). *Let $A : [a, b] \to \mathbb{R}$ and $B : [a, b] \to \mathbb{R}$ be two functions. If the function $\Phi : [a, b] \to \mathbb{R}$ satisfies the recursive differential inequality*

$$\Phi'(x) \leq A(x)\Phi(x) + B(x), \qquad \forall x \in [a, b], \tag{12}$$

*then it also satisfies the inequality*

$$\Phi(x) \leq \Phi(a) \exp\left\{ \int_a^x A(z)\, dz \right\} + \int_a^x B(s) \exp\left\{ \int_s^x A(z)\, dz \right\} ds, \qquad \forall x \in [a, b].$$

*Proof.* To ease notation, we set $E(x) = \exp\{-\int_a^x A(z)\, dz\}$. By multiplying both sides of (12) by $E(x)$, we get

$$\left( \Phi(x)E(x) \right)' \leq B(x)E(x), \qquad \forall x \in [a, b].$$

Integrating this inequality, we arrive at

$$\Phi(x)E(x) \leq \Phi(a)E(a) + \int_a^x B(s)E(s)\, ds.$$

Dividing both sides of this inequality by $E(x) > 0$ and taking into account that $E(a) = 1$, we get the claim of the lemma. $\qquad\square$

**Lemma 5** (Integral form). *Let $A : [a, b] \to [0, +\infty)$ and $B : [a, b] \to \mathbb{R}$ be two functions. If the function $\Phi : [a, b] \to \mathbb{R}$ satisfies the recursive integral inequality*

$$\Phi(x) \leq \int_a^x A(s)\Phi(s)\, ds + B(x), \qquad \forall x \in [a, b],$$

*then it also satisfies the inequality*

$$\Phi(x) \leq \int_a^x A(s)B(s) \exp\left\{ \int_s^x A(z)\, dz \right\} ds + B(x), \qquad \forall x \in [a, b]. \tag{13}$$

*Proof.* We set

$$\Psi(x) = \exp\left\{ -\int_a^x A(z)\, dz \right\} \int_a^x A(s)\Phi(s)\, ds.$$

We have

$$\Psi'(x) = -A(x)\Psi(x) + \exp\left\{ -\int_a^x A(z)\, dz \right\} A(x)\Phi(x)$$

$$\leq -A(x)\Psi(x) + \exp\left\{ -\int_a^x A(z)\, dz \right\} A(x)\left( \int_a^x A(s)\Phi(s)\, ds + B(x) \right)$$

$$= -A(x)\Psi(x) + A(x)\Psi(x) + A(x)B(x) \exp\left\{ -\int_a^x A(z)\, dz \right\}.$$

Therefore,

$$\Psi(x) \leq \Psi(a) + \int_a^x A(s)B(s) \exp\left\{ -\int_a^s A(z)\, dz \right\} ds.$$

Replacing $\Psi$ by its expression and using the fact that $\Psi(a) = 0$, we get

$$\exp\left\{ -\int_a^x A(z)\, dz \right\} \int_a^x A(s)\Phi(s)\, ds \leq \int_a^x A(s)B(s) \exp\left\{ -\int_a^s A(z)\, dz \right\} ds.$$

This implies that

$$\int_a^x A(s)\Phi(s)\, ds \leq \int_a^x A(s)B(s) \exp\left\{ \int_s^x A(z)\, dz \right\} ds.$$

Combining this inequality with (13), we get the claim of the lemma. $\qquad\square$

# B    Proof of Proposition 1

For the penalty factor $\alpha(t) = 1/(A + 2t)$, we get $\beta(t) = \int_0^t \alpha(s)\,ds = (1/2) \log\left(1 + (2/A)t\right)$. This implies that

$$\sqrt{\mu_2(\pi)}\,e^{-\beta(t)} + 11\mu_2(\pi)\sqrt{\alpha(t)} = \frac{\sqrt{A\mu_2(\pi)} + 11\mu_2(\pi)}{\sqrt{A + 2t}}.$$

Finally, the middle term in the right hand side of (2) takes the form

$$11\mu_2(\pi) \int_0^t \frac{|\alpha'(s)|}{\sqrt{\alpha(s)}} e^{\beta(s)-\beta(t)} ds = \frac{11\mu_2(\pi)}{\sqrt{A + 2t}} \int_0^t \frac{2}{A + 2s} ds$$

$$= \frac{11\mu_2(\pi)}{\sqrt{A + 2t}} \log\left(1 + (2/A)t\right).$$

Combining these relations, we get the claim of the proposition.

# C    (Weakly) convex potentials: what is known and what we can hope for

Many recent papers investigated the case of strongly convex potential; this case is now rather well understood. Let us briefly summarize here some facts and conjectures that can shed some light on the broader case of weakly convex potential. This might help to understand what can be expected to be proved in the framework s<tudied in this work.

The ergodicity properties of the Langevin process are closely related to such notions of functional analysis as the spectral gap, the Poincaré and the log-Sobolev inequalities. Thus, the generator of a Markov semi-group associated with an $m$-strongly convex potential has a spectral-gap $\mathcal{C}_{\mathrm{SG}}$ at least equal to $m$. This property was exploited by Dalalyan (2017) to derive guarantees on the LMC algorithm. It is known that the spectral gap exists if and only if the invariant density satisfies the Poincaré inequality. Furthermore, the spectral gap is equal to the inverse of the Poincaré constant $\mathcal{C}_{\mathrm{P}}$. Furthermore, distributions associated to $m$-strongly convex potentials satisfy the log-Sobolev inequality with the constant $\mathcal{C}_{\mathrm{LS}} \leq 1/m$. This property was used by Durmus and Moulines (2019) to extend the guarantees to the Wasserstein-2 distance.

Note that the log-Sobolev inequality is stronger than the Poincaré inequality and $\mathcal{C}_{\mathrm{P}} \leq \mathcal{C}_{\mathrm{LS}}$. For $m$-strongly convex potentials, we have $\mathcal{C}_{\mathrm{SG}}^{-1} = \mathcal{C}_{\mathrm{P}} \leq \mathcal{C}_{\mathrm{LS}} \leq 1/m$. Results in (Dalalyan, 2017; Durmus and Moulines, 2019) imply that in order to get a Wasserstein distance smaller than $\varepsilon\sqrt{p/m}$, it suffices to perform a number of LMC iterations proportional to $(M/m)^2 \varepsilon^{-2}$, up to logarithmic factors. A formal proof of the fact that the same result holds for the densities satisfying the log-Sobolev inequality with constant $1/m$ (but which are not necessarily $m$-strongly log-concave) was given in (Vempala and Wibisono, 2019).

On the other hand, it was established by (Bobkov, 1999) that any log-concave distribution satisfies the Poincaré inequality. However, the Poincaré constant might depend on the dimension. In (Kannan et al., 1995), the authors conjectured that there is a universal constant $\mathcal{C}_{\mathrm{KLS}} > 0$ such that for any log-concave distribution $\pi$ on $\mathbb{R}^p$,

$$\mathcal{C}_{\mathrm{P}} \leq \mathcal{C}_{\mathrm{KLS}} \|\mathbf{E}_\pi[\boldsymbol{X}\boldsymbol{X}^\top]\|_{\mathrm{op}} := \mathcal{C}_{\mathrm{KLS}}\mu_{\mathrm{op}}(\pi). \tag{KLS}$$

Despite important efforts made in recent years (see (Alonso-Gutiérrez and Bastero, 2015; Cattiaux and Guillin, 2018)), this conjecture is still unproved. Finally, in the recent paper (Chewi et al., 2020), Corollary 4 establishes that $W_2(\mu_t^{\mathsf{LD}}, \pi) \leq \sqrt{2\mathcal{C}_{\mathrm{P}}\chi^2(\nu_0\|\pi)}\,e^{-t/\mathcal{C}_{\mathrm{P}}}$. While the exponential in $t$ convergence to zero is a very appealing property of this result, it comes with two shortcomings. To the previously mentioned difficulty of assessing the Poincaré constant, one has to add the challenging problem of finding a meaningful upper bound on the $\chi^2$-divergence between the initial distribution and the target.

What can we hope for in the light of the previous discussion? As shown in (Dalalyan, 2017, Lemma 5), for $f$ satisfying $(m, M)$-SCGL, choosing $\nu_0 = \mathcal{N}(\boldsymbol{x}_*, M^{-1}\mathbf{I}_p)$ yields $\chi^2(\nu_0\|\pi) \leq (M/m)^{p/2}$. In the case $m = 0$, it might be possible to replace $m$ by $1/\mathcal{C}_{\mathrm{P}}$ in this result. If in addition, we admit

inequality KLS, then we get

$$W_2^2(\mu_t^{\mathsf{LD}}, \pi) \le 2\mathcal{C}_{\mathsf{P}} \left(M\mathcal{C}_{\mathsf{P}}\right)^{p/2} e^{-2t/\mathcal{C}_{\mathsf{P}}}$$

$$\le 2\mathcal{C}_{\mathsf{KLS}}\mu_{\mathrm{op}}(\pi) \left(M\mathcal{C}_{\mathsf{KLS}}\mu_{\mathrm{op}}(\pi)\right)^{p/2} e^{-2t/\mathcal{C}_{\mathsf{KLS}}\mu_{\mathrm{op}}(\pi)}.$$

This is, probably, the best upper bound one could hope for in the general log-concave setting by Langevin diffusion based algorithms. We see that it has three drawbacks as compared to our result stated in Proposition 1. First, it requires the knowledge of a minimizer $\boldsymbol{x}^*$. Second, it involves the Lipschitz constant $M$ of the gradient. Third, it is heavily based on $\mathcal{C}_{\mathsf{KLS}}$, which might be very large.

## D Penalized Gradient Flow

### D.1 Proof of Theorem 2

We recall that for every $\gamma \in \mathbb{R}$, is given by $f_\gamma(\cdot) := f(\cdot) + \gamma\|\cdot\|_2^2/2$. We define $\boldsymbol{x}_\gamma$ the minimum point of $f_\gamma$. In particular, $\boldsymbol{x}_0 = \boldsymbol{x}_*$. The triangle inequality yields

$$\|\boldsymbol{X}_t^{\mathsf{PGF}} - \boldsymbol{x}_*\|_2 \le \|\boldsymbol{X}_t^{\mathsf{PGF}} - \boldsymbol{x}_{\alpha(t)}\|_2 + \|\boldsymbol{x}_{\alpha(t)} - \boldsymbol{x}_0\|_2 \qquad (14)$$

for every $t > 0$. We will bound these two terms separately. $\mathbf{A}(\mathsf{A}, \mathsf{q})$ for $\gamma = 0$ and $\widetilde{\gamma} = \alpha(t)$ yields the following bound on the second term:

$$\|\boldsymbol{x}_{\alpha(t)} - \boldsymbol{x}_0\|_2 \le \mathsf{D}\alpha(t)^{1-\mathsf{q}}\|\boldsymbol{x}_*\|^{1-\mathsf{q}}.$$

To bound the first term of (14), we aim at obtaining a Gronwall-type inequality for the function

$$\phi(t) := \|\boldsymbol{X}_t^{\mathsf{PGF}} - \boldsymbol{x}_{\alpha(t)}\|_2.$$

To this end, we consider an auxiliary stochastic process $\{\widetilde{\boldsymbol{X}}_u : u \ge t\}$, defined as a solution of the following differential equation

$$d\widetilde{\boldsymbol{X}}_u = -\left(\nabla f(\widetilde{\boldsymbol{X}}_u) + \alpha(t)\widetilde{\boldsymbol{X}}_u\right)du,$$

with the starting point $\widetilde{\boldsymbol{X}}_t = \boldsymbol{X}_t$. This is in fact the gradient flow corresponding to the strongly-convex potential $f_{\alpha(t)}$. The triangle inequality yields

$$\phi(t + \delta) \le \left\|\boldsymbol{X}_{t+\delta}^{\mathsf{PGF}} - \widetilde{\boldsymbol{X}}_{t+\delta}\right\|_2 + \left\|\widetilde{\boldsymbol{X}}_{t+\delta} - \boldsymbol{x}_{\alpha(t)}\right\|_2 + \left\|\boldsymbol{x}_{\alpha(t)} - \boldsymbol{x}_{\alpha(t+\delta)}\right\|_2.$$

From the linear convergence of the gradient flow of an $\alpha(t)$-strongly convex function, we get the following:

$$\left\|\boldsymbol{X}_{t+\delta}^{\mathsf{PGF}} - \boldsymbol{x}_{\alpha(t)}\right\|_2 \le \exp\left(-\delta\alpha(t)\right)\left\|\widetilde{\boldsymbol{X}}_t - \boldsymbol{x}_{\alpha(t)}\right\|_2 = \exp\left(-\delta\alpha(t)\right)\phi(t).$$

In order to bound the distance between $\boldsymbol{x}_{\alpha(t)}$ and $\boldsymbol{x}_{\alpha(t+\delta)}$, we use again $\mathbf{A}(\mathsf{A}, \mathsf{q})$ condition, thus

$$\left\|\boldsymbol{x}_{\alpha(t)} - \boldsymbol{x}_{\alpha(t+\delta)}\right\|_2 \le \frac{\mathsf{D}}{\alpha^{\mathsf{q}}(t)}(\alpha(t) - \alpha(t+\delta))\|\boldsymbol{x}_*\|_2^{1-\mathsf{q}}.$$

Thus we obtain a bound for $\phi(t + \delta)$, that depends linearly on $\phi(t)$:

$$\phi(t + \delta) \le \left\|\boldsymbol{X}_{t+\delta}^{\mathsf{PGF}} - \widetilde{\boldsymbol{X}}_{t+\delta}\right\|_2 + e^{-\delta\alpha(t)}\phi(t) + \frac{\mathsf{D}}{\alpha^{\mathsf{q}}(t)}(\alpha(t) - \alpha(t+\delta))\|\boldsymbol{x}_*\|_2^{1-\mathsf{q}}. \qquad (15)$$

Let us subtract $\phi(t)$ from both sides of (15) and divide by $\delta$:

$$\frac{\phi(t + \delta) - \phi(t)}{\delta} \le \frac{1}{\delta} \cdot \left\|\boldsymbol{X}_{t+\delta}^{\mathsf{PGF}} - \widetilde{\boldsymbol{X}}_{t+\delta}\right\|_2 + \frac{\exp\left(-\delta\alpha(t)\right) - 1}{\delta} \cdot \phi(t) \qquad (16)$$

$$+ \frac{\mathsf{D}(\alpha(t) - \alpha(t+\delta))}{\delta\alpha^{\mathsf{q}}(t)}\|\boldsymbol{x}_*\|_2^{1-\mathsf{q}}.$$

The next lemma provides an upper bound on $\left\|\boldsymbol{X}_{t+\delta}^{\mathsf{PGF}} - \widetilde{\boldsymbol{X}}_{t+\delta}\right\|_2$ showing that it is $o(\delta)$, when $\delta \to 0$.

**Lemma 6.** *Suppose $f$ satisfies $(m, M)$-SCGL with $m = 0$. Then for every $t, \delta > 0$, and for every integrable function $\alpha : [t, t + \delta] \to [0, \infty)$,*

$$\|\widetilde{\boldsymbol{X}}_{t+\delta} - \boldsymbol{X}^{\mathsf{PGF}}_{t+\delta}\|_2 \leq \left(\phi(t) + \|\boldsymbol{x}_{\alpha(t)}\|_2\right) \exp\left\{M\delta + \int_0^\delta \alpha(t+u)\, du\right\} \int_0^\delta \left|\alpha(t+s) - \alpha(t)\right| ds.$$

The proof can be found in the Appendix D.2. When $\delta$ tends to 0, according to Lemma 6, the first term of the right-hand side of (16) vanishes. Thus, after passing to the limit, we are left with the following Gronwall-type inequality:

$$\phi'(t) \leq -\alpha(t)\phi(t) - \frac{\mathsf{D}\alpha'(t)}{\alpha^{\mathsf{q}}(t)} \cdot \|\boldsymbol{x}_*\|_2^{1-\mathsf{q}}. \tag{17}$$

Here we tacitly used the fact that $\|\boldsymbol{x}_{\alpha(t+\delta)}\|_2 \leq \|\boldsymbol{x}_0\|_2$. Recalling that the function $\beta(t)$ is given by $\beta(t) = \int_0^t \alpha(s)ds$, one can rewrite (17) as

$$\left(\phi(t)e^{\beta(t)}\right)' \leq -\frac{\mathsf{D}\alpha'(t)e^{\beta(t)}}{\alpha^{\mathsf{q}}(t)}\|\boldsymbol{x}_*\|_2^{1-\mathsf{q}}.$$

Therefore we infer the following bound on $\phi(t)$:

$$\phi(t) \leq \phi(0)e^{-\beta(t)} - \mathsf{D}\|\boldsymbol{x}_*\|_2^{1-\mathsf{q}} \int_0^t \frac{\alpha'(s)}{\alpha^{\mathsf{q}}(s)}e^{\beta(s)-\beta(t)}ds.$$

Combining this bound with (7), we obtain the inequality

$$\|\boldsymbol{X}^{\mathsf{PGF}}_t - \boldsymbol{x}_0\|_2 \leq \|\boldsymbol{X}^{\mathsf{PGF}}_0 - \boldsymbol{x}_{\alpha(0)}\|_2 e^{-\beta(t)} - \mathsf{D}\|\boldsymbol{x}_*\|_2^{1-\mathsf{q}} \int_0^t \frac{\alpha'(s)}{\alpha^{\mathsf{q}}(s)}e^{\beta(s)-\beta(t)}ds + \mathsf{D}\|\boldsymbol{x}_*\|_2^{1-\mathsf{q}}\alpha(t)^{1-\mathsf{q}}.$$

Since the process $\boldsymbol{X}^{\mathsf{PGF}}_t$ starts at point 0, $\|\boldsymbol{X}^{\mathsf{PGF}}_0 - \boldsymbol{x}_{\alpha(0)}\|_2 = \|\boldsymbol{x}_{\alpha(0)}\|_2$. The next lemma bounds $\|\boldsymbol{x}_{\alpha(0)}\|_2$.

**Lemma 7.** *The function $\gamma \mapsto \|\boldsymbol{x}_\gamma\|_2$ is a non-increasing continuous function on the interval $[0, \infty)$.*

Therefore, $\|\boldsymbol{x}_{\alpha(0)}\|_2 \leq \|\boldsymbol{x}_0\|_2 = \|\boldsymbol{x}_*\|_2$, which completes the proof of Theorem 2.

### D.2 Proof of Lemma 6

From the definition of $\widetilde{\boldsymbol{X}}$, we can write

$$\widetilde{\boldsymbol{X}}_{t+\delta} - \boldsymbol{X}^{\mathsf{PGF}}_{t+\delta} = \int_t^{t+\delta} \left(\nabla f(\boldsymbol{X}^{\mathsf{PGF}}_s) - \nabla f(\widetilde{\boldsymbol{X}}_s) + \alpha(s)\boldsymbol{X}^{\mathsf{PGF}}_s - \alpha(t)\widetilde{\boldsymbol{X}}_s\right) ds.$$

Therefore we have

$$\|\widetilde{\boldsymbol{X}}_{t+\delta} - \boldsymbol{X}^{\mathsf{PGF}}_{t+\delta}\|_2 \leq \underbrace{\left\|\int_t^{t+\delta} \left(\nabla f(\boldsymbol{X}^{\mathsf{PGF}}_s) - \nabla f(\widetilde{\boldsymbol{X}}_s)\right) ds\right\|_2}_{:=T_1} + \underbrace{\left\|\int_t^{t+\delta} \left(\alpha(s)\boldsymbol{X}^{\mathsf{PGF}}_s - \alpha(t)\widetilde{\boldsymbol{X}}_s\right) ds\right\|_2}_{:=T_2}.$$

Now let us analyze these two terms separately. We start with $T_1$:

$$\begin{aligned}
\|T_1\|_2 &= \left\|\int_t^{t+\delta} \left(\nabla f(\boldsymbol{X}^{\mathsf{PGF}}_s) - \nabla f(\widetilde{\boldsymbol{X}}_s)\right) ds\right\|_2 \\
&\leq \int_t^{t+\delta} \left\|\nabla f(\boldsymbol{X}^{\mathsf{PGF}}_s) - \nabla f(\widetilde{\boldsymbol{X}}_s)\right\|_2 ds \\
&\leq M \int_t^{t+\delta} \|\boldsymbol{X}^{\mathsf{PGF}}_s - \widetilde{\boldsymbol{X}}_s\|_2 ds.
\end{aligned}$$

These are due to the Minkowskii inequality and the Lipschitz continuity of the gradient. In order to bound the second term $T_2$, we will add and subtract the term $\alpha(t+s)\widetilde{\boldsymbol{X}}_{t+s}$. Similar to the case above, we get the following upper bound:

$$\|T_2\|_2 \leq \int_t^{t+\delta} \alpha(s) \left\|\boldsymbol{X}_s^{\mathsf{PGF}} - \widetilde{\boldsymbol{X}}_s\right\|_2 ds + \int_t^{t+\delta} |\alpha(s) - \alpha(t)| \left\|\widetilde{\boldsymbol{X}}_s\right\|_2 ds$$

$$= \int_0^\delta \alpha(t+s) \left\|\boldsymbol{X}_{t+s}^{\mathsf{PGF}} - \widetilde{\boldsymbol{X}}_{t+s}\right\|_2 ds + \int_0^\delta |\alpha(t+s) - \alpha(t)| \left\|\widetilde{\boldsymbol{X}}_{t+s}\right\|_2 ds.$$

Recall that $\widetilde{\boldsymbol{X}}_{t+s}$ is the gradient flow of an $(m+\alpha(t))$-strongly convex potential function. Thus, the triangle inequality yields

$$\left\|\widetilde{\boldsymbol{X}}_{t+s}\right\|_2 \leq \left\|\widetilde{\boldsymbol{X}}_{t+s} - \boldsymbol{x}(t)\right\|_2 + \|\boldsymbol{x}(t)\|_2$$

$$\leq \left\|\widetilde{\boldsymbol{X}}_t - \boldsymbol{x}_t\right\|_2 \exp(-ms - \alpha(t)s) + \|\boldsymbol{x}_t\|_2$$

$$\leq \left\|\boldsymbol{X}_t^{\mathsf{PGF}} - \boldsymbol{x}_t\right\|_2 + \|\boldsymbol{x}_t\|_2 := V_t.$$

Summing up, we have

$$\left\|\boldsymbol{X}_{t+\delta}^{\mathsf{PGF}} - \widetilde{\boldsymbol{X}}_{t+\delta}\right\|_2 \leq \int_0^\delta \left(M + \alpha(t+s)\right) \|\boldsymbol{X}_{t+s}^{\mathsf{PGF}} - \widetilde{\boldsymbol{X}}_{t+s}\|_2 ds + \widetilde{\alpha}_t(\delta)\, V_t,$$

where $\widetilde{\alpha}_t(\delta)$ is an auxiliary function defined as

$$\widetilde{\alpha}_t(\delta) := \int_0^\delta |\alpha(t+s) - \alpha(t)|\, ds.$$

Now let us define $\Phi(s)L := \|\boldsymbol{X}_{t+s}^{\mathsf{PGF}} - \widetilde{\boldsymbol{X}}_{t+s}\|_2$. The last inequality can be rewritten as

$$\Phi(\delta) \leq \int_0^\delta \left(M + \alpha(t+s)\right)\Phi(s)\, ds + \widetilde{\alpha}_t(\delta)\, V_t.$$

The (integral form of the) Gronwall inequality implies that

$$\Phi(\delta) \leq V_t \int_0^\delta \widetilde{\alpha}_t(s)\left(M + \alpha(t+s)\right)e^{\int_s^\delta (M+\alpha(t+u))\,du}\, ds + \widetilde{\alpha}_t(\delta) V_t$$

$$= V_t \int_0^\delta \widetilde{\alpha}_t'(s)\, e^{\int_s^\delta (M+\alpha(t+u))\,du}\, ds$$

$$\leq V_t\, \widetilde{\alpha}_t(\delta)\, \exp\left\{M\delta + \int_0^\delta \alpha(t+u)\, du\right\}.$$

This completes the proof.

### D.3    Proof of Lemma 7

Suppose that $\gamma_1 < \gamma_2$. We want to show that $\|\boldsymbol{x}_{\gamma_1}\|_2 > \|\boldsymbol{x}_{\gamma_2}\|_2$. Let us consider the function $f_{\gamma_2}$. We have that

$$f_{\gamma_2}(\boldsymbol{x}_{\gamma_2}) \leq f_{\gamma_2}(\boldsymbol{x}_{\gamma_1}) = f(\boldsymbol{x}_{\gamma_1}) + \gamma_2\|\boldsymbol{x}_{\gamma_1}\|_2/2.$$

The definition of $f_{\gamma_1}$ yields

$$f_{\gamma_2}(\boldsymbol{x}_{\gamma_2}) \leq f_{\gamma_1}(\boldsymbol{x}_{\gamma_1}) + (\gamma_2 - \gamma_1)\|\boldsymbol{x}_{\gamma_1}\|_2/2$$

$$\leq f_{\gamma_1}(\boldsymbol{x}_{\gamma_2}) + (\gamma_2 - \gamma_1)\|\boldsymbol{x}_{\gamma_1}\|_2/2$$

$$\leq f_{\gamma_2}(\boldsymbol{x}_{\gamma_2}) + (\gamma_2 - \gamma_1)\left(\|\boldsymbol{x}_{\gamma_1}\|_2 - \|\boldsymbol{x}_{\gamma_2}\|_2\right)/2.$$

Here the second passage is valid, as $\boldsymbol{x}_{\gamma_1}$ is the minimum point of $f_{\gamma_1}$. Since $\gamma_2 > \gamma_1$, the difference $\|\boldsymbol{x}_{\gamma_1}\|_2 - \|\boldsymbol{x}_{\gamma_2}\|_2$ is positive. Thus the monotony is proved.

To prove the continuity of the function we take a sequence $\gamma_n$ that tends to $\gamma_0$ and show that $\boldsymbol{x}_{\gamma_n} \to \boldsymbol{x}_{\gamma_0}$. Assumption A(D, q) yields

$$\|\boldsymbol{x}_{\gamma_n} - \boldsymbol{x}_{\gamma_0}\|_2 \leq \frac{\mathsf{D}}{\max(\gamma_n, \gamma_0)^{\mathsf{q}}} |\gamma_n - \gamma_0| \|\boldsymbol{x}_*\|_2, \qquad \forall n \in \mathbb{N}.$$

Since $\mathsf{q} < 1$, the ratio of $|\gamma_n - \gamma_0|$ and $\max(\gamma_n, \gamma_0)^{\mathsf{q}}$ tends to zero, when $n \to 0$. This concludes the proof.

# E  Examples of functions satisfying condition $\mathbf{A}(\mathsf{D}, \mathsf{q})$

In this section we consider several functions that are convex but not strongly convex and satisfy $\mathbf{A}(\mathsf{D}, \mathsf{q})$ condition presented in Section 3.

## E.1  Locally strongly convex functions

We prove that locally strongly convex functions satisfy $\mathbf{A}(\mathsf{D}, 0)$. Recalling Lemma 7 we get that $\|\boldsymbol{x}_\gamma\|_2 \le \|\boldsymbol{x}_*\|_2$. Thus the we can consider the function only on $\mathcal{B}(0, \|x_*\|_2)$. Since $f$ is locally strongly convex, there exists $m_*$ such that it is $m_*$-strongly convex in the ball $\mathcal{B}(0, \|x_*\|_2)$. The latter means, that $f_{\widetilde\gamma}$ is $(m_* + \widetilde\gamma)$-strongly convex. Therefore (Nesterov, 2004)[Theorem 2.1.9] yields the following:

$$\|\boldsymbol{x}_\gamma - \boldsymbol{x}_{\widetilde\gamma}\|_2 \le \frac{1}{m_* + \widetilde\gamma}\|\nabla f_{\widetilde\gamma}(\boldsymbol{x}_\gamma) - \nabla f_{\widetilde\gamma}(\boldsymbol{x}_{\widetilde\gamma})\|_2.$$

Using the optimality condition for differentiable functions one gets $\nabla f_{\widetilde\gamma}(\boldsymbol{x}_\gamma) = (\widetilde\gamma - \gamma)\boldsymbol{x}_\gamma$ for all $\gamma \ge 0$. Therefore, for every $0 \le \gamma < \widetilde\gamma$, we obtain

$$\|\boldsymbol{x}_\gamma - \boldsymbol{x}_{\widetilde\gamma}\|_2 \le \frac{1}{m_* + \widetilde\gamma}\|(\widetilde\gamma - \gamma)\boldsymbol{x}_\gamma\|_2 \le \frac{\widetilde\gamma - \gamma}{m_*}\|\boldsymbol{x}_\gamma\|_2.$$

The latter is true due to Lemma 7. Thus $f$ satisfies $\mathbf{A}(1/m_*, 0)$.

## E.2  Cubic function $f(\boldsymbol{x}) = \|\boldsymbol{x} - \boldsymbol{x}_*\|_2^3$

In this section we show that the cubic function satisfies $\mathbf{A}(1/\sqrt{3\|\boldsymbol{x}_*\|_2}, 1/2)$. It is straightforward to verify that the function $f$ is convex. $f_\gamma$ is strongly convex and the optimality condition for $\boldsymbol{x}_\gamma$ yields the following equality:

$$\nabla f(\boldsymbol{x}_\gamma) + \gamma\boldsymbol{x}_\gamma = 3\|\boldsymbol{x}_\gamma - \boldsymbol{x}_*\|_2(\boldsymbol{x}_\gamma - \boldsymbol{x}_*) + \gamma\boldsymbol{x}_\gamma = 0. \tag{18}$$

In the case when $\boldsymbol{x}_* = 0$, the penalized minimum point $\boldsymbol{x}_\gamma$ equals 0, for every $\gamma$, thus we suppose in the following that $\boldsymbol{x}_* \ne 0$. Since the norm is scalar, (18) yields that the vectors $\boldsymbol{x}_\gamma - \boldsymbol{x}_*$ and $\boldsymbol{x}_\gamma$ are co-linear. Therefore there exists a real number $\lambda_\gamma$ such that $\boldsymbol{x}_\gamma = \lambda_\gamma\boldsymbol{x}_*$. Lemma 7 implies that $|\lambda_\gamma| \le 1$, thus the following quadratic equality is true:

$$-3\|\boldsymbol{x}_*\|_2(\lambda_\gamma - 1)^2\boldsymbol{x}_* + \gamma\lambda_\gamma\boldsymbol{x}_* = 0. \tag{19}$$

As said in the beginning, $\boldsymbol{x}_* \ne 0$, therefore it its coefficient that is equal to zero. Solving the quadratic equation with respect to $\lambda_\gamma$, we get the following formula:

$$\lambda_\gamma = 1 - \frac{\gamma}{\gamma/2 + \sqrt{3\gamma\|\boldsymbol{x}_*\|_2 + \gamma^2/4}}.$$

According to Lemma 7, for every $\widetilde\gamma > \gamma$, we have $|\lambda_\gamma| > |\lambda_{\widetilde\gamma}|$. On the other hand, from (19) one deduces that $\lambda_\gamma > 0$, for every $\gamma > 0$. Thus, inserting the found value for $\lambda_\gamma$, we obtain the following inequality:

$$\|\boldsymbol{x}_\gamma - \boldsymbol{x}_{\widetilde\gamma}\|_2 = \|\boldsymbol{x}_*\|_2 \left( \frac{\widetilde\gamma}{\widetilde\gamma/2 + \sqrt{3\widetilde\gamma\|\boldsymbol{x}_*\|_2 + \widetilde\gamma^2/4}} - \frac{\gamma}{\gamma/2 + \sqrt{3\gamma\|\boldsymbol{x}_*\|_2 + \gamma^2/4}} \right)$$

$$\le \frac{(\widetilde\gamma - \gamma)\|\boldsymbol{x}_*\|_2}{\widetilde\gamma/2 + \sqrt{3\widetilde\gamma\|\boldsymbol{x}_*\|_2 + \widetilde\gamma^2/4}} \le \frac{(\widetilde\gamma - \gamma)\|\boldsymbol{x}_*\|_2^{1/2}}{\sqrt{3\widetilde\gamma}}.$$

Therefore $f$ satisfies $\mathbf{A}(1/\sqrt{3}, 1/2)$.

## E.3  Power function $f(\boldsymbol{x}) = \|\boldsymbol{x} - \boldsymbol{x}_*\|_2^a$

For $a \ge 2$, we consider the function $f(\boldsymbol{x}) = \|\boldsymbol{x} - \boldsymbol{x}_*\|_2^a$. We show here that $f$ satisfies $\mathbf{A}((1/a)^{1/(a-1)}, (a-2)/(a-1))$. Since $f_\gamma$ is a differentiable strongly-convex function, we get the following equation for $\boldsymbol{x}_\gamma$:

$$a\|\boldsymbol{x}_\gamma - \boldsymbol{x}_*\|_2^{a-2}(\boldsymbol{x}_\gamma - \boldsymbol{x}_*) + \gamma\boldsymbol{x}_\gamma = 0. \tag{20}$$

Similar to the previous case, we notice that $\boldsymbol{x}_\gamma - \boldsymbol{x}_*$ and $\boldsymbol{x}_\gamma$ are co-linear. Thus, there exists $\lambda_\gamma$ such that $\boldsymbol{x}_\gamma = (1 - \lambda_\gamma)\boldsymbol{x}_*$. Since $\boldsymbol{x}_*$ is assumed to be non-zero, in order to calculate $\boldsymbol{x}_\gamma$, one needs to solve the following equation:

$$|\lambda_\gamma|^{a-2}\lambda_\gamma = \frac{\gamma(1 - \lambda_\gamma)}{a\|\boldsymbol{x}_*\|^{a-2}}. \tag{21}$$

Thus the $p$-dimensional equation (20) reduces to equation (21) involving a one-dimensional unknown. Lemma 7 yields $\lambda_{\widetilde{\gamma}} > \lambda_\gamma > 0$ for every $\widetilde{\gamma} > \gamma \geq 0$. In addition, from (21), we have that $\lambda_\gamma \leq 1$ for every $\gamma > 0$. It is straightforward to verify that for every $\gamma \geq 0$, (21) has exactly one solution satisfying these conditions.

**Lemma 8.** *Let $\alpha \geq 1$. If $(\lambda_s : s \in (0, 1))$ satisfies $\lambda_s^\alpha = s(1 - \lambda_s)$ for every $s \in (0, 1)$, then*

$$|\lambda_s - \lambda_{s'}| \leq \frac{|s - s'|}{(s \vee s')^{(\alpha-1)/\alpha}}, \qquad \forall s', s \in (0, 1).$$

*Proof.* Without loss of generality, we assume that $s' \leq s$. Computing the derivative of both sides of the identity $\lambda_s^\alpha = s(1 - \lambda_s)$, we get

$$\lambda_s' = \frac{1 - \lambda_s}{\alpha\lambda_s^{\alpha-1} + s} \geq 0.$$

This implies that $\lambda_{s'} \leq \lambda_s$. In addition,

$$\begin{aligned}
\lambda_s - \lambda_{s'} &\leq \frac{\lambda_s^\alpha - \lambda_{s'}^\alpha}{\lambda_s^{\alpha-1}} \\
&= \frac{s(1 - \lambda_s) - s'(1 - \lambda_{s'})}{\lambda_s^{\alpha-1}} \\
&= \frac{(s - s')(1 - \lambda_{s'})}{\lambda_s^{\alpha-1}} - \frac{s(\lambda_s - \lambda_{s'})}{\lambda_s^{\alpha-1}}.
\end{aligned}$$

Rearranging the terms, we arrive at

$$\begin{aligned}
\lambda_s - \lambda_{s'} &\leq \frac{(s - s')(1 - \lambda_{s'})}{\lambda_s^{\alpha-1}\left(1 + \frac{s}{\lambda_s^{\alpha-1}}\right)} \\
&= \frac{(s - s')(1 - \lambda_{s'})}{\lambda_s^{\alpha-1} + s}
\end{aligned}$$

In the last fraction, the numerator is bounded by $s - s'$, while the denominator satisfies

$$\begin{aligned}
\lambda_s^{\alpha-1} + s &= (s(1 - \lambda_s))^{(\alpha-1)/\alpha} + s \\
&\geq (s(1 - s^{1/\alpha}))^{(\alpha-1)/\alpha} + s \\
&\geq s^{(\alpha-1)/\alpha}(1 - s^{1/\alpha}) + s = s^{(\alpha-1)/\alpha}.
\end{aligned}$$

This completes the proof of the lemma. $\qquad\square$

Applying this lemma to (21), we get

$$\lambda_{\widetilde{\gamma}} - \lambda_\gamma \leq \frac{\widetilde{\gamma} - \gamma}{a\|\boldsymbol{x}_*\|_2^{a-2}(\widetilde{\gamma}/a\|\boldsymbol{x}_*\|_2^{a-2})^{(a-2)/(a-1)}} = \frac{\widetilde{\gamma} - \gamma}{a^{1/(a-1)}\|\boldsymbol{x}_*\|_2^{(a-2)/(a-1)}\widetilde{\gamma}^{(a-2)/(a-1)}},$$

for all $\gamma, \widetilde{\gamma}$ satisfying $0 \leq \gamma \leq \widetilde{\gamma} \leq a\|\boldsymbol{x}_*\|_2^{a-2}$. In conclusion, we get

$$\begin{aligned}
\|\boldsymbol{x}_{\widetilde{\gamma}} - \boldsymbol{x}_\gamma\|_2 &\leq \frac{\widetilde{\gamma} - \gamma}{a^{1/(a-1)}\|\boldsymbol{x}_*\|_2^{(a-2)/(a-1)}\widetilde{\gamma}^{(a-2)/(a-1)}}\|\boldsymbol{x}_*\|_2 \\
&\leq \frac{\widetilde{\gamma} - \gamma}{\widetilde{\gamma}^{(a-2)/(a-1)}}(\|\boldsymbol{x}_*\|_2/a)^{1/(a-1)}.
\end{aligned}$$

This concludes the proof.