[Reviews · NeurIPS 2020]

Review 1

Summary and Contributions: The paper studies the problem of sampling from a log concave and smooth distribution. The authors introduce new continuous dynamics, the penalized Langevin dynamics for which they provide an upper bound on the mixing time. Then they consider the repercussion of these new dynamics in optimization by taking the low temperature limit.

Strengths: The problem of sampling from a log concave distribution has received a lot of attention within the NeurIPS community and is of great importance with major applications in diverse domains (such as Bayesian inference). The majority of these works had focused on the simplest strongly log concave case where the potential is strongly convex. In this setting, convergence proofs can be adapted from convex optimization without insurmontable difficulties. The case of weakly log concave distribution (where the potential is convex but not strongly convex) has been, in comparison, neglected. However, in practice, problems are often not strongly log concave or even if they are, the strong convexity constant can be arbitrarily small. Therefore, the studied problem in this paper is of primary interest for the NeurIPS community.

Weaknesses: The contributions of this paper are unfortunately too limited for a top conference like NeurIPS. The main contribution of the paper is to extend previous works on penalized Langevin to time-dependent penalization parameters. However, the authors do not carry over the analysis of the discretization error (which is let as future work in the conclusion). On the other side, this extension does not seem to be very challenging and the proof is classical. It amounts to using the triangular inequality to express the distance between the penalized dynamics and the target distribution as the sum of the distance between the penalized dynamics and the penalized target distribution and the distance between the target distribution and the penalized target distribution. Since the potential of the penalized dynamics is strongly convex, classical results can be applied and the distance between the penalized and unpenalized target distributions is controlled using a transportation cost inequality. The analysis made after theorem 1 is also interesting but not very challenging. The contribution on the penalized gradient flow is interesting but not really relevant, in the sense that Theorem 2 appears to be more an interesting curiosity rather than answering an important question in optimization.

Correctness: The claims and the method are correct.

Clarity: The paper is very nicely written and pleasant to read. Concepts are very clearly explained and the authors are very precise with the mathematical notations and tools. Theorems are well commented. It shows perfect mastery of the topics. This was the best written papers of the ones I had to review this year.

Relation to Prior Work: The related work is very well done. - l40-41: recent works of Lee and Vempala could be cited too. - l 43-45: references of works where it is done could be interesting (if they exist).

Reproducibility: Yes

Additional Feedback: To conclude, this paper is very nicely written and pleasant to read and try to answer an important and overlooked question. However, its contributions are too incremental as is. It would really benefit from results on the discretization of the dynamics and mixing time for the implementable algorithm to be further compared to existing approaches. Then it would be a very nice submission to NeurIPS. Minor comments: - l140: this assumption should be further discussed. ________________________ After reading the authors's rebuttal, I have decided to keep my overall score unchanged.


Review 2

Summary and Contributions: This paper studies a version of Langevin dynamics with a time varying penalty (a time varying quadratic is added to the potential) added to deal with the problem of sampling from weakly log-concave distributions. They also show an interesting result that characterizes the distance diffusion process and a minimizer of the potential in the low temperature limit.

Strengths: The paper is well-written and it very easy to read. The authors do an excellent job in motivating their problem and relating it to past work. The idea of considering a penalized gradient Langevin dynamics is clean and intuitive. The proofs have been carefully executed and are easy to read and understand. The results are also quite interesting and I am sure this paper will be valuable addition to the growing literature on this problem of sampling from log-concave distribution using Langevin-based algorithms. The authors also do a good job in discussing the implications of their theoretical results.

Weaknesses: One obvious weakness is that the author do not analyze a discretized version of their algorithm which could be algorithmically implementable. It feels like this should be an straightforward to do and would be a valuable addition to the paper.

Correctness: I verified most of the proofs in the main paper and supplementary material and they seem correct to me.

Clarity: The paper is written to a high standard and is easy to understand.

Relation to Prior Work: The authors do an excellent job of relating to past work on this problem. They also discuss the implications of their theoretical results well.

Reproducibility: Yes

Additional Feedback: Questions: 1. Does diffusion process end up closest to minimizer (x^*) that is closest to the origin since the penalty is the gradient of a time varying quadratic centered at the origin? 2. In the setting where the potential is weakly log-concave (m=0), do you think a discretization of this diffusion process (with a time varying penalty) will be faster, in terms of number of iterations as compared to simple Langevin Monte Carlo (with a fixed quadratic penalty) added? A discussion regarding this in the paper might be valuable. ++++++ Post author feedback ++++++ After discussion with the other reviewers I agree that without an analysis of the discretization error it is hard to judge what the time-varying penalization buys over previous processes and algorithms. For this reason I have reduced my score.


Review 3

Summary and Contributions: They introduce a new diffusion process derived from Langevin Dynamics for log-concave measures. The process is a time-inhomogenuous Markov process that consists of adding a time dependent drift term so that "local" stationnary measure is strongly local convave. It is a rather natural idea given the good behavior of Langevin Dynamics for strongly loc-concave measures. They obtain non asymptotic rates of convergences in Wasserstein of the law of the process to the limit measure. They derive an optimazation counterpart of this process and derive similar rates of convergence.

Strengths: The paper introduces a natural yet interesting variant of the Langevin Dynamics and provide a bound for this process. The application to the optimization seem to lead to a better rate of convergence compared to the ones previously known, for the gradient flow of convex functions.

Weaknesses: The rate of convergence is much slower than the original Langevin Dynamics: log-concave measures enjoy exponential rates of convergence while this new process seems to have a linear rate of convergence. They obtain an "optimal" rate of convergence by choosinga particular alpha that decreases in t; however, choosing alpha=0 gives a faster rate of convergence, suggesting that their upper bound is loose. There is no numerical experiment that compare the rates of convergence with classical Langevin Dynamics in the cases where the assumptions are true. The assumption A, although discussed in the paper, remains a bit mysterious to me for the interesting case of q<1.

Correctness: The claims seem correct to me, although I did not check into details all the proofs in the appendix.

Clarity: The paper is well written and easy to follow. The assumptions are simple and clear.

Relation to Prior Work: While the paper cite a lot of previous work, they do not explicitely compare their results. Even though rates of convergence in continuous time do not tell much about discretize schemes rates of convergence, it would still have been interesting to compare the rates of convergence of the different time-continuous variants of gradient flow (such as Nesterov accelartion) for the optimization and variants of Langevin Dynamics for the sampling. Similarily, numerical experiments comparing the two classical Langevin Dynamics and the proposed penalized one would have shed a light on the practical side of this ideas.

Reproducibility: Yes

Additional Feedback: There is a typo in the main theorem. There should be a minus sign before the beta in the exponent in the second term of the RHS of equation (2) ===== Post rebuttal ===== After reading the author's rebuttal and a discussion with other reviewers, I keep the same score. While I am okay with the study focusing on continuous time, I am still not sure about the optimality of the upper bound, while I value a deeper discussion on assumption A. ====================


Review 4

Summary and Contributions: The authors propose a time-inhomogeneous SDE based on Langevin dynamics they call penalized Langevin dynamics (PLD) and posit a bound between the 2-Wasserstein metric of the invariant measure pi and marginal distribution pi_t. They also purpose a similar type of bound that relates to gradient flows as opposed to SDEs.

Strengths: The paper contains some interesting theoretical results for understanding how to schedule the change in temperature during an sampling or optimization procedure. In particular, for the sampling case, they provide a new upper bound on the 2-Wasserstein metric which decays at rate O(t^{-1/2}). Moreover, by rescaling the PLD dynamics, the authors provide some subsequent results that bound the distance of minimizer and the current value as a function of time (also making Assumption A).

Weaknesses: The biggest concern I have about the paper is whether these results are indeed novel. Langevin dynamics have been modified in many ways, and I myself and not positive whether similar results have already been proved in the literature. Moreover, it looks like the bound on the Wasserstein-2 metric is O(t^{-1/2}) for m=0, while it appears that exponentially decreasing bounds exist (Chewi et al. 2020). The authors explain in the appendix that their bound doesn't require unknown constants (e.g. the Poincare constant) but it does require knowing mu_2(pi). Of course this can be estimated, but its unclear practically how much better this bound is than the exponentially fast converging one with some other constants that must be estimated.

Correctness: The results seem plausible and the proofs in the appendix look reasonable as well.

Clarity: Yes, the paper is well written.

Relation to Prior Work: The authors do discuss the relationship to other lines of work to prove the convergence of Langevin dynamics to the invariant distribution in the case of m=0, but they do not really discuss other work on temperature-related augmentations to the dynamics. I am a bit concerned that similar results may already exist.

Reproducibility: Yes

Additional Feedback: ==== Post-rebuttal ==== After reading the other reviewers' questions and comments and the author rebuttal, I have decided to stick with my previous assessment. The authors did a satisfactory job answers my concerns, but I also am not confident enough to raise my score based on the other authors' discussion.

[Author Response · NeurIPS 2020]

While other Neurips authors complain massively on social networks about the quality of the reports on their papers, we consider ourselves privileged having received 4 fair and constructive reports. We would like to wholeheartedly thank the reviewers for their suggestions and thought provoking remarks. We really want to present our work at NeurIPS and hope that our responses below will convince the reviewers that this submission is worth being accepted.

**Answers to Reviewer # 1**

1. [. . . ] authors do not carry over the analysis of the discretization error. We find it impossible to add the results on discretization to this paper without sacrificing the clarity and the precision, and respecting the 9-pages limitation. There are many different discretization schemes: Euler, Ozaki, randomized mid-point, *etc*. Each of these schemes has its strengths and weaknesses. We are preparing a separate paper analyzing and comparing these discretization schemes along with two novel variants. Note that the randomized mid-point discretization alone was the object of a full NeurIPS paper [2] with 18 pages supplementary material. We believe that an in-depth study of continuous-time dynamics is of interest on its own (this opinion seems to be shared by the authors of [3,4]).

2. [. . . ] this extension does not seem to be very challenging and the proof is classical.
   We disagree with the reviewer on this assessment. As kindly mentioned by the reviewer in other comments, the paper is written in a mathematically detailed manner and every passage has been meticulously polished. This is perhaps the main reason why the proof, as it is currently presented, seems not challenging and classical. On the other hand, the term "classical" is not appropriate here, since we are not aware of any other paper where such a proof is used.

**Answers to Reviewer # 2**

3. [. . . ] Does diffusion process end up closest to minimizer that is closest to the origin? Yes, this is easy to prove. We will mention this point along with a sketch of proof in the revised version.

4. [. . . ] In the setting $m = 0$, do you think a discretization of this diffusion process will be faster, in terms of number of iterations as compared to simple LMC with a fixed quadratic penalty added? According to our preliminary rough computations, using a clever discretization leads to rates that improve on results in [1]. Let us also mention here that an obvious advantage of the time-varying penalty is the adaptivity to the time horizon.

**Answers to Reviewer # 3**

5. They obtain an "optimal" rate of convergence by choosing a particular alpha that decreases in t; however, choosing alpha=0 gives a faster rate of convergence, suggesting that their upper bound is loose.
   The upper bound depends of course on $t$, but also on other parameters such as the dimension and the condition number. The best known [4] exponential bound is $W_2^2(\nu_t, \pi) \leq 2\mathcal{C}_P e^{-2t/\mathcal{C}_P} \chi^2(\nu_0, \pi)$. The $\chi^2$-divergence therein might be very large (recall that for $m$-strongly convex potentials the $\mu_2$ is of order $p/m$ while the $\chi^2$-divergence is $(M/m)^p$ [Lemma 5, arXiv:1412.7392]). The currently available upper bounds on $\mathcal{C}_P$ depend also badly on $p$. All in all, the dependence of our bound on $p$ and the condition number is much better than that of the foregoing exponential-in-$t$ bound. This makes it advantageous to use our bound in the high-dimensional or badly conditioned settings. We will further elaborate on this in the revised version.

6. The assumption A, although discussed in the paper, remains a bit mysterious to me for the interesting case of $q < 1$.
   We agree and intend to further discuss this assumption in the revised version.

7. [. . . ] do not explicitly compare their results. Even though rates in continuous time do not tell much about discretize schemes rates, it would still have been interesting to compare the rates of convergence of the different time-continuous variants of gradient flow (such as Nesterov accelartion) for the optimization and variants of Langevin Dynamics for the sampling. Thank you for this suggestion. We will revise the paper accordingly if accepted.

**Answers to Reviewer # 4**

8. [. . . ] it looks like the bound on the Wasserstein-2 metric is $O(t^{-1/2})$ for $m = 0$, while it appears that exponentially decreasing bounds exist (Chewi et al. 2020). Please see our response to Q5 above (raised by Rev #3).

9. The authors explain in the appendix that their bound doesn't require unknown constants (e.g. the Poincare constant) but it does require knowing $\mu_2(\pi)$ [. . . ]. Estimating the second-order moment (SOM) is arguably qualitatively simpler than estimating the Poincaré constant (PC). But this is not the main advantage of our result. For many concrete weakly convex potentials, one can easily compute (see Prop 2-4 in [1]) upper bounds on the SOM in which the dependence on the dimension is tight and polynomial. This is not the case of the PC. Finally, our result leads to an upper bound which is smaller than the so-far-known exponential-in-$t$ bound for a large range of values of $t$.

10. [. . . ] do not really discuss other work on temperature-related augmentations to the dynamics. I am a bit concerned that similar results may already exist. We would be grateful if the reviewer could provide some references.

**References**

[1] Dalalyan, Karagulyan and Riou-Durand (2020) arXiv:1906.08530.

[2] Shen and Lee (2019) NeurIPS and arXiv:1909.05503.

[3] Su, Boyd and Candes (2014) NIPS and arXiv:1503.01243.

[4] Chewi, S., Gouic, T. L., Lu, C., Maunu, T., Rigollet, P., and Stromme, A. (2020) arXiv:2005.09669.


[Meta-Review · NeurIPS 2020]

The paper was heavily discussed post-rebuttals. All the reviewers appreciate the clean mathematical analysis and writing, though on a technical level, the analysis is quite similar to prior work. However, the lack of discretization analysis was considered a serious flaw by all reviewers -- as it's unclear what are the algorithmic implications for the problem studied (furthermore, given prior work on discretizing Langevin, everyone felt this wouldn't be too hard and wouldn't harm the clarity that much).